# Mfn2 ubiquitination by PINK1/parkin gates the p97-dependent release of ER from mitochondria to drive mitophagy

**Gian-Luca McLelland[1,2†], Thomas Goiran[1,2], Wei Yi[1,2], Geneviève Dorval[1,2,3], Carol X Chen[1,2,3], Nadine D Lauinger[1,2,3], Andrea I Krahn[1,2], Sepideh Valimehr[4], Aleksandar Rakovic[5], Isabelle Rouiller[4], Thomas M Durcan[1,2,3], Jean-François Trempe[6], Edward A Fon[1,2,3]***

[1]McGill Parkinson Program, Montreal Neurological Institute, McGill University, Montreal, Canada; [2]Neurodegenerative Diseases Group, Montreal Neurological Institute, McGill University, Montreal, Canada; [3]iPSC-CRISPR Platform, Montreal Neurological Institute, McGill University, Montreal, Canada; [4]Department of Anatomy & Cell Biology, McGill University, Montreal, Canada; [5]Institute of Neurogenetics, University of Lübeck, Lübeck, Germany; [6]Department of Pharmacology & Therapeutics, McGill University, Montreal, Canada

**\*For correspondence:**
ted.fon@mcgill.ca

**Present address:** [†]Netherlands Cancer Institute, Amsterdam, Netherlands

**Competing interests:** The authors declare that no competing interests exist.

**Abstract** Despite their importance as signaling hubs, the function of mitochondria-ER contact sites in mitochondrial quality control pathways remains unexplored. Here we describe a mechanism by which Mfn2, a mitochondria-ER tether, gates the autophagic turnover of mitochondria by PINK1 and parkin. Mitochondria-ER appositions are destroyed during mitophagy, and reducing mitochondria-ER contacts increases the rate of mitochondrial degradation. Mechanistically, parkin/ PINK1 catalyze a rapid burst of Mfn2 phosphoubiquitination to trigger p97-dependent disassembly of Mfn2 complexes from the outer mitochondrial membrane, dissociating mitochondria from the ER. We additionally demonstrate that a major portion of the facilitatory effect of p97 on mitophagy is epistatic to Mfn2 and promotes the availability of other parkin substrates such as VDAC1. Finally, we reconstitute the action of these factors on Mfn2 and VDAC1 ubiquitination in a cell-free assay. We show that mitochondria-ER tethering suppresses mitophagy and describe a parkin-/PINK1-dependent mechanism that regulates the destruction of mitochondria-ER contact sites.
DOI: https://doi.org/10.7554/eLife.32866.001

## Introduction

Loss of *PRKN* or *PINK1* results in an early-onset form of hereditary Parkinson's disease (PD), a neurological disorder that is linked to mitochondrial dysfunction (*Kitada et al., 1998*; *Ryan et al., 2015*; *Valente et al., 2004*). Accordingly, parkin and PINK1 promote mitochondrial health through several mitochondrial quality control mechanisms; the turnover of outer mitochondrial membrane (OMM) proteins by the proteasome, the generation of mitochondrial-derived vesicles, and whole-organellar degradation by mitophagy, a form of selective autophagy (*Sugiura et al., 2014*; *Yamano et al., 2016*). During mitophagy, PINK1, a mitochondrial kinase, builds up on the surface of damaged mitochondria where it activates parkin directly via phosphorylation and allosterically through the generation of phosphoubiquitin (pUb) (*Kane et al., 2014*; *Kazlauskaite et al., 2014*; *Kondapalli et al., 2012*; *Koyano et al., 2014*; *Shiba-Fukushima et al., 2012*). Parkin, an E3 ubiquitin (Ub) ligase, mediates the ubiquitination of resident OMM proteins, recruiting Ub-binding autophagic machinery through a feed-forward mechanism to ultimately degrade the organelle via the lysosome (*Heo et al., 2015*; *Lazarou et al., 2015*; *Ordureau et al., 2015*; *Ordureau et al., 2014*).

Contact sites between mitochondria and the endoplasmic reticulum (ER) act as crucial signaling hubs in the context of non-selective, starvation-induced autophagy, where they serve as the site of autophagosome formation (*Hamasaki et al., 2013*; *Kishi-Itakura et al., 2014*). Indeed, autophagosome biogenesis is impaired in cells with defective mitochondria-ER tethering (*Hamasaki et al., 2013*), as lipid transfer between organelles may be important for their formation (*Hailey et al., 2010*; *Klecker et al., 2014*). As steady-state mitophagy in yeast requires mitochondria-ER contacts (*Böckler and Westermann, 2014*), it has been assumed that parkin-dependent mitophagy follows a similar mechanism (*Yoshii and Mizushima, 2015*). However, this model directly conflicts with the observation that mitofusin-2 (Mfn2) – a mitochondria-ER tether required for starvation-induced autophagosome formation in mammals (*de Brito and Scorrano, 2008*; *Hamasaki et al., 2013*; *Naon et al., 2016*) – is ubiquitinated by parkin and rapidly turned over by the proteasome (*Tanaka et al., 2010*). Thus, how mitophagy is regulated by contacts between mitochondria and the ER (if at all), and the location from which the mitophagic membrane originates, remain open questions in the field.

## Results

### Parkin and PINK1 destroy mitochondria-ER contact during mitophagy

We hypothesized that PINK1 and parkin may regulate contact between both organelles during mitophagy, based on studies demonstrating high levels of parkin ubiquitination activity on Mfn2 in both cells and *in organello* ubiquitination assays (*Tanaka et al., 2010*; *Tang et al., 2017*). To first determine whether parkin destroys the OMM-ER interface of depolarized mitochondria, we analyzed contacts between the two organelles by electron microscopy (EM) (*Csordás et al., 2006*). We quantified ER tubules within 100 nm of the OMM, as this distance is enough to capture tubules closely associated with the OMM (*Figure 1A*, left panel and inset). To induce PINK1-/parkin-mediated mitophagy, we treated U2OS cells stably-expressing GFP-parkin (U2OS:GFP-parkin) and control U2OS:GFP cells with CCCP for four hours, and observed by EM a decrease the total length of ER-OMM contact in both cell lines, although this decrease was greater in magnitude in cells expressing GFP-parkin (*Figure 1A*, quantified in 1B). However, when CCCP-induced, parkin-independent mitochondrial fragmentation was taken into account (*Figure 1C*), parkin had a specific effect on reducing the percentage of the OMM that remained in contact with the ER in depolarized cells (*Figure 1D*), as well as the percentage of total mitochondria that were still connected to the ER (*Figure 1E*). This effect was robust, as repeating our quantification using a variety of interorganellar tethering lengths – ER-OMM distances of 100, 50 and 25 nm (*Figure 1—figure supplement 1A and B*) – pointed us to the same conclusion; parkin disrupts mitochondria-ER contact upon activation of mitophagy. Indeed, this effect was indiscriminate in that it was not selective for one subset of ER-OMM distances (*Figure 1—figure supplement 1C*). Moreover, the subsets of remaining contacts observed after the ~75% reduction in CCCP-treated, GFP-parkin-expressing cells (*Figure 1D* and *Figure 1—figure supplement 1C*) were biased towards longer interorganellar distances (*Figure 1—figure supplement 1D*), consistent with parkin driving the OMM and ER apart. Given that the mitochondria observed in our EM analyses were still intact organelles and not yet engulfed by the isolation membrane (IM) of the autophagosome (*Figure 1A*, right panel), we concluded that parkin ablates contact between mitochondria and the ER as an early step during depolarization-induced mitophagy in cells.

We next took a closer look at how this process of contact site removal may occur (for the remainder of our study, we used the <100 nm interorganellar distance to quantify ER-OMM contacts). Parkin has been reported, through its ability to ubiquitinate OMM proteins and target them for proteasomal degradation, to eventually mediate the rupture of the OMM prior to or during engulfment by the autophagosome (*Yoshii et al., 2011*). Indeed, we observed rare (likely transient) mitochondrial structures where we believed OMM rupture to be occurring at the time of fixation (*Figure 1F*, the blue arrowheads indicate the limits of OMM rupture, where the organelle is being wrapped by the IM [indicated by the broken green line]). Concordantly, ER contacts with the still-intact OMM were observed (*Figure 1F*, red arrowheads), leading us to postulate that the removal of OMM-ER contacts may precede OMM rupture. To this end, we quantified ER-OMM contacts in CCCP-treated cells that were co-incubated with the proteasome inhibitor MG132, which stabilizes the unmodified band of OMM parkin substrates, including Mfn2, and prevents rupture of the OMM

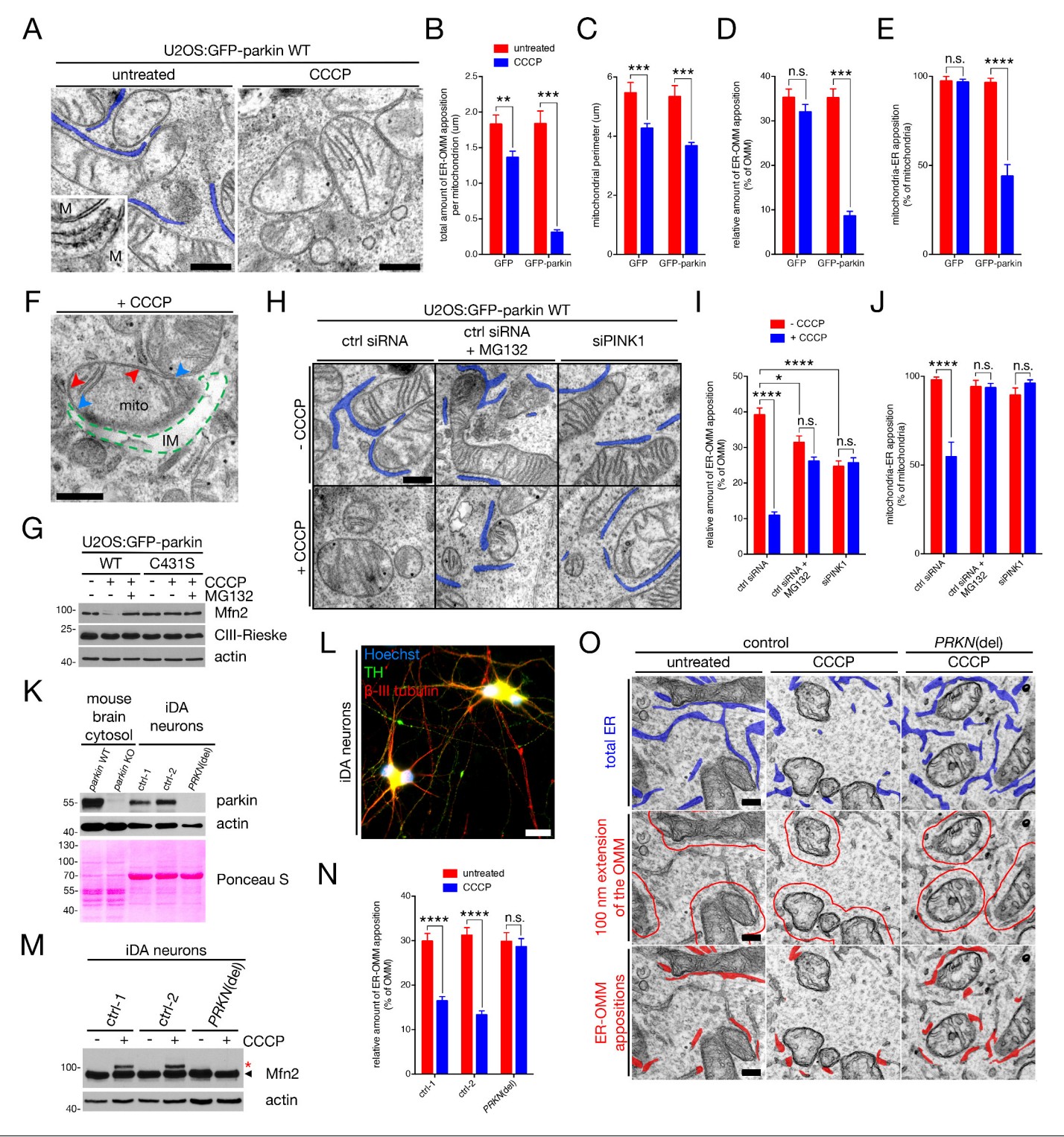

**Figure 1.** Ultrastructural analysis of ER-mitochondria contact during mitophagy in U2OS cells and dopaminergic neurons. (**A**) Representative TEM images of mitochondria ('M') in contact with ER (pseudocoloured blue) in untreated and CCCP-treated U2OS:GFP-parkin cells. Scale bars, 500 nm. (**B–E**) Quantification of TEM from (**A**) in U2OS:GFP and GFP-parkin WT cells, left untreated (red bars) or treated with 20 μM CCCP for four hours (blue bars). Total apposition length (**B**), mitochondrial size (**C**), and the percent of OMM per mitochondrion (**D**) and mitochondria per field (**E**) in contact with the ER was quantified. Bars represent mean ± SEM, n = 82 to 152 mitochondria in 15 to 19 fields per condition. n.s., not significant; **, p<0.01; ***, p<0.001; ****, p<0.0001. (**F**) TEM image of an isolation membrane ('IM', broken green line) wrapping a mitochondrion ('mito'). Blue arrowheads indicate
*Figure 1 continued on next page*

*Figure 1 continued*

the boundaries of OMM rupture, while red arrowheads indicate ER tubules in contact with the intact portion of the OMM. Scale bar, 500 nm. (**G**) Immunoblot analysis of whole-cell lysates from U2OS:GFP-parkin WT and C431S cells treated with 20 µM CCCP for four hours with or without 10 µM MG132. In the case of MG132 treatment, cells were first pre-incubated with 10 µM MG132 for 30 min prior to addition of CCCP. (**H**) Representative TEM images of mitochondria in contact with ER (pseudocoloured blue) in U2OS:GFP-parkin WT cells transfected with the indicated siRNA, and treated with 20 µM CCCP ('+CCCP') for four hours, in the presence or absence of 10 µM MG132 as in (**G**). Scale bar, 500 nm. (**I,J**) Quantification of TEM from (**H**) in cells treated with (blue bars) or without (red bars) 20 µM CCCP for four hours. The percent of OMM per mitochondrion (**I**) and mitochondria per field (**J**) in contact with the ER were quantified. Bars represent mean ± SEM, n = 101 to 203 mitochondria in 14 to 16 fields per condition. n.s., not significant; *, p<0.05; ***, p<0.001; ****, p<0.0001. (**K**) Immunoblot analysis of parkin levels in mouse brain cytosol from parkin$^{+/+}$ and parkin$^{-/-}$ mice, along with whole-cell lysates from iDA neurons derived from iPSCs isolated from control (ctrl) individuals and a *PRKN* patient ("*PRKN*(del)"). (**L**) A representative wide-field image showing that iDA neurons express TH (green) and β-III tubulin (red) (Hoechst, blue). Scale bar, 20 microns. (**M**) Immunoblot analysis of whole-cell lysates from iDA neurons treated with 20 µM CCCP for one hour. The arrowhead indicates the unmodified Mfn2 band, while the red asterisk indicates ubiquitinated Mfn2. (**N**) Quantification of the percent of the OMM opposed to the ER in iDA neurons treated with 20 µM CCCP for one hour. Bars represent mean ± SEM, n = 80 to 131 mitochondria per condition. n.s., not significant; ****, p<0.0001. (**O**) Representative TEM images of mitochondria in contact with ER in iDA neurons. In the top row, total ER is pseudocoloured blue. In the second row, the red line denotes an area within 100 nm of the OMM. In the bottom row, ER tubules within the 100 nm area are pseudocoloured red. Scale bars, 200 nm.

DOI: https://doi.org/10.7554/eLife.32866.002

The following source data and figure supplement are available for figure 1:

**Source data 1.** Numerical source data for *Figure 1B–1D, E, I, J and N* and *Figure 1—figure supplement 1B to D*.
DOI: https://doi.org/10.7554/eLife.32866.004
**Figure supplement 1.** Mitophagy reduces ER-OMM contacts of all intermembrane distances.
DOI: https://doi.org/10.7554/eLife.32866.003

(*Chan et al., 2011*; *Rakovic et al., 2011*; *Yoshii et al., 2011*) (*Figure 1G*, GFP-parkin$^{C431S}$, which cannot ligate Ub (*Trempe et al., 2013*), is used as a negative control). MG132 co-incubation rescued ER-OMM contact in U2OS:GFP-parkin cells treated with CCCP (*Figure 1H,I and J*). As expected, we also prevented OMM-ER disruption in cells depleted of PINK1 (*Figure 1H,I and J*).

Finally, we replicated our U2OS cell data in induced pluripotent stem cell (iPSC) -derived dopaminergic (iDA) neurons isolated from either control individuals or a patient carrying compound heterozygous deletions in the *PRKN* gene (*PRKN*$^{del}$; see Materials and methods). iDA neuronal cultures express endogenous parkin at a level comparable to that in the cytosolic fraction from mouse brain (*Figure 1K*), as well as the catecholinergic marker tyrosine hydroxylase (TH) (*Figure 1L*). Full-length parkin was undetectable in *PRKN*$^{del}$ cells (*Figure 1K*), as expected given the genetic background of this line (*Grünewald et al., 2010*). Upon treatment of these neurons with CCCP for only one hour, we observed Mfn2 ubiquitination in both control lines but not in the parkin deletion line (*Figure 1M*). When we analyzed mitochondria-ER appositions in these cells, we again observed a CCCP-dependent decrease in the amount of <100 nm ER-OMM appositions in both control lines (*Figure 1N and O*). However, this decrease was absent in the parkin deletion line (*Figure 1N and O*), supporting our previous overexpression data in U2OS cells (*Figure 1A to E* ). Thus, PINK1 and parkin function to destroy contacts between the ER and mitochondria during mitophagy, likely through parkin-mediated OMM protein ubiquitination and turnover, as this process can be prevented by inhibiting proteasomal degradation. Moreover, this is a relevant biological process in human dopamine neurons, where it is regulated by endogenous parkin.

## Phosphoubiquitination of Mfn2 by the PINK1/parkin system disrupts its antagonistic effect on mitophagy

Our EM data demonstrated that ER-mitochondria uncoupling occurs as an early step in the mitophagy pathway, prior to autophagosomal engulfment of the organelle (*Figure 1F*), and we sought to understand the underlying mechanism of this phenomenon. Mfn2 is both a mitochondria-ER tether and parkin ubiquitination substrate (*de Brito and Scorrano, 2008*; *Sarraf et al., 2013*; *Tanaka et al., 2010*), and thus the modulation of interorganellar contact by PINK1/parkin may occur through their effect on Mfn2. We began by examining the ubiquitination (via the disappearance of the unmodified band) of various parkin substrates (*Khan et al., 2016*; *Sarraf et al., 2013*) during a CCCP time course in U2OS:GFP-parkin cells, using the A320R mutant – which fails to bind pUb and initiate mitophagy (*Wauer et al., 2015a*; *Yamano et al., 2015*) – as a negative control. Turnover of

both Mfn1 and Mfn2 occurred early (almost complete disappearance by two hours) compared to other OMM proteins (*Figure 2A*). Upon higher exposure (*Figure 2B*) of these immunoblots (from *Figure 2A*), we observed a rapid 'burst' of Mfn2 ubiquitination that occurred between 30 and 60 min CCCP. When compared to TOM20, a protein that is not promptly ubiquitinated by parkin (*Sarraf et al., 2013*), the rapidity of this Ub burst on Mfn2 was emphasized as TOM20 ubiquitination occurs gradually over a period of hours, rather than rapidly over a period of minutes (*Figure 2B*). Thus, ubiquitination of the mitofusins is one of the very first steps after the induction of mitophagy.

Mechanistically, this Ub burst would require local activation of parkin by PINK1 in the vicinity of Mfn2, which could be achieved by PINK1-catalyzed phosphorylation of the resulting Ub chains – events that would dually serve to activate parkin and tether it in place (*Okatsu et al., 2015*). To test this, we first immunoprecipitated WT or A320R GFP-parkin from cells treated with CCCP over time. We observed robust coimmunoprecipitation of ubiquitinated Mfn1 and Mfn2 with GFP-parkin$^{WT}$ at one hour CCCP (corresponding to the Ub burst observed in *Figure 2B*), with no apparent binding at four hours (*Figure 2C*), likely due to turnover of the Mfns by the proteasome at this time (*Figures 1G* and *2B* and [*Tanaka et al., 2010*]). When we analyzed other parkin substrates that are ubiquitinated less rapidly than the Mfns (*Figure 2A*), we observed binding to WT parkin only at four hours of CCCP treatment in the case of ubiquitinated Miro1, and binding of mono-ubiquitinated HK1 at one hour CCCP, which was further shifted at four hours, indicative of processivity of HK1 ubiquitination (*Figure 2C*). None of these ubiquitinated species coimmunoprecipitated with GFP-parkin$^{A320R}$ (*Figure 2C*). To confirm that GFP-parkin was indeed binding ubiquitinated Mfn2, we treated GFP-parkin immunoprecipitates from CCCP-treated cells with Usp2 deubiquitinase (see schematic in *Figure 2D*), which is active on both phosphorylated and unphosphorylated Ub chains (*Wauer et al., 2015b*), and observed the release of Mfn2 from the parkin-bound bead fraction into the supernatant after separation by centrifugation (*Figure 2E*). These results strongly suggested that, early on in the mitophagy pathway, parkin was binding ubiquitinated Mfn2, likely through interactions with pUb moieties.

We next confirmed the phosphoubiquitination of Mfn2 during mitochondrial depolarization. When we immunoprecipitated Mfn2 from U2OS:GFP-parkin$^{WT}$ cells that were treated with CCCP for one hour, we detected Ub-modified species by immunoblot (*Figure 2F*). This was concomitant with a decrease in overall Mfn2 levels (*Figure 2F*), owing to its proteasomal turnover (*Figure 1G*). Liquid-chromatography coupled to mass spectrometry (LC/MS) confirmed that the Mfn2 immunoprecipitation contained pS65 Ub selectively in the CCCP-treated condition (*Figure 2G*), despite lower Mfn2 levels (*Figure 2A and F* and *Figure 2—figure supplement 1*). We then confirmed that both pS65 and unphosphorylated Ub were covalently attached to Mfn2 by its precipitation under denaturing conditions and detecting pS65 Ub and total Ub by immunoblot (*Figure 2H*). Finally, profiting from the nanomolar affinity of the parkin R0RBR module for pS65 Ub (*Sauvé et al., 2015*), we used GST-R0RBR to pull down phosphoubiquitinated species from CCCP-treated U2OS:GFP-parkin$^{WT}$ cell lysates. We again used the A320R mutant – which abolishes the parkin-pUb interaction (*Figure 2I*) (*Wauer et al., 2015a*; *Yamano et al., 2015*) – as a negative control. In a CCCP-dependent manner, pS65 Ub, Ub and (shifted) Mfn2 could be detected in GST-R0RBR$^{WT}$ pulldowns (*Figure 2J*). Strikingly, we did not observe any of these factors in pulldowns using GST-R0RBR$^{A320R}$ (*Figure 2J*). Mfn2 is therefore phosphoubiquitinated and, taken together with our previous data, a burst of phosphoubiquitination – parkin-mediated ubiquitination coupled to PINK1-catalyzed phosphorylation – occurs on Mfn2 at an early time point in the mitophagy pathway.

Our observations so far demonstrated that mitochondria are separated from the ER during mitophagy, and that the OMM-ER tether Mfn2 is rapidly degraded at the onset of the pathway. We thus hypothesized that Mfn2 may antagonize mitophagy through its ability to tether mitochondria and the ER, necessitating its destruction. To test this, we silenced Mfn2 (siMfn2) in U2OS:GFP-parkin$^{WT}$ cells, as well as Mfn1 – which promotes mitochondrial fusion without any apparent role in interorganellar tethering (*de Brito and Scorrano, 2008*) – to control for phenomena resulting from fusion defects. We confirmed Mfn1 and Mfn2 depletion by immunoblot (*Figure 3A*), and observed mitochondrial fragmentation in both siMfn1 and siMfn2 cells (*Figure 3B* and *Figure 3—figure supplement 1A and B*) with an ER-OMM apposition defect unique to the siMfn2 condition (*Figure 3—figure supplement 1A,C and D*), as expected. Next, we investigated the kinetics of parkin recruitment to depolarized mitochondria in these cells (in our analyses, a cell is considered to have recruited parkin if the parkin signal covers the mitochondrial reticulum in its entirety). Moreover, we

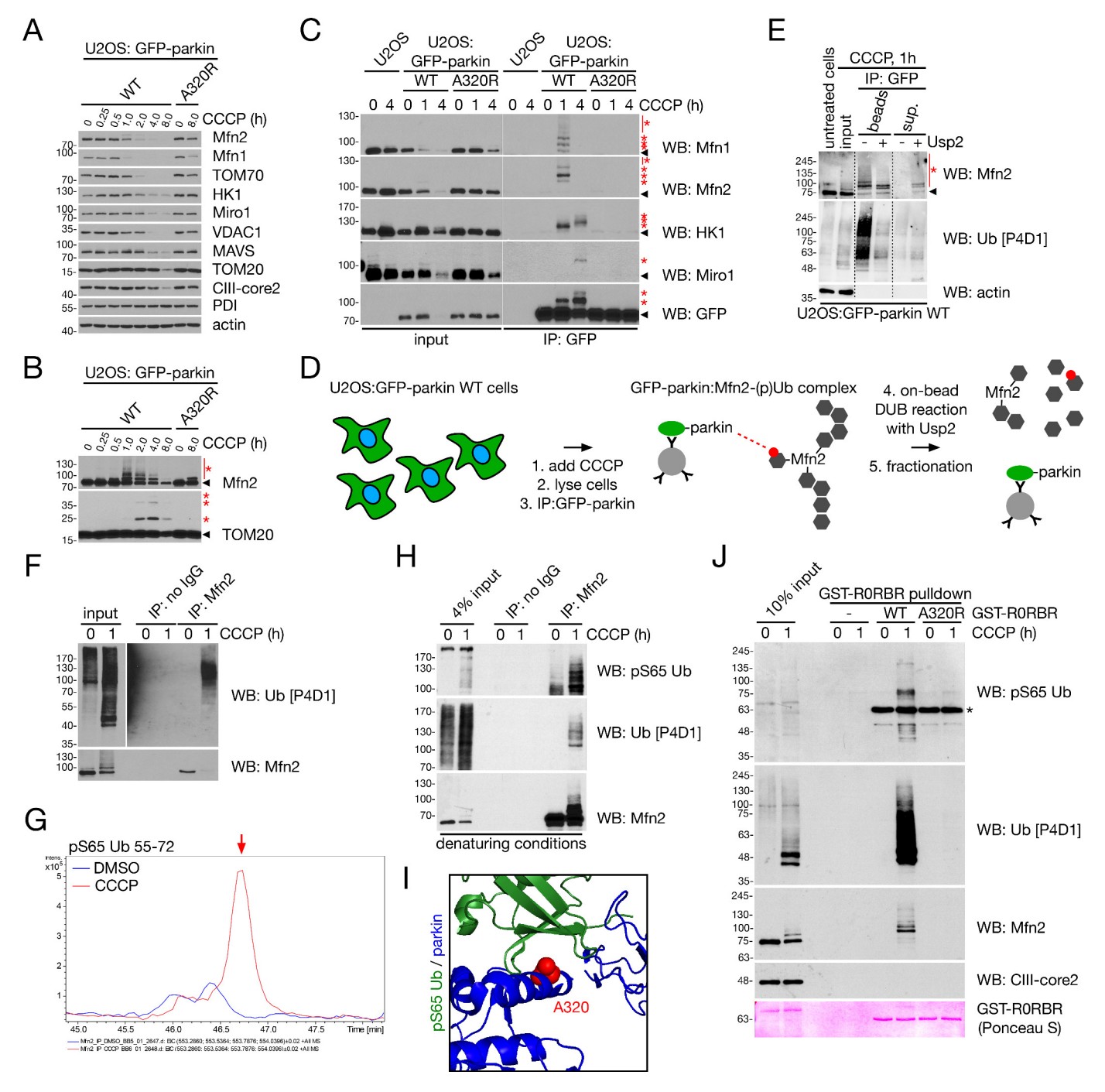

**Figure 2.** Mfn2 is rapidly phosphoubiquitinated upon induction of mitophagy. (**A**) Immunoblot analysis of protein turnover in glucose-maintained U2OS: GFP-parkin WT and A320R cells treated with 20 μM CCCP for the indicated time. (**B**) Higher exposures of Mfn2 and TOM20 immunoblots from (**A**). Red asterisks indicate ubiquitinated forms of Mfn2 and TOM20. (**C**) Co-immunoprecipitation of parkin substrates with GFP-parkin WT or A320R in U2OS cells treated with 20 μM CCCP for the indicated time, using an anti-GFP antibody. Immunoprecipitates were separated, along with 4% input, by SDS-PAGE and immunoblotted for the indicated protein. The arrowhead indicates the unmodified form of the protein, while the red asterisks denote ubiquitinated forms. (**D**) Workflow for the on-bead deubiquitination of Mfn2. U2OS:GFP-parkin WT cells were treated for one hour with 20 μM CCCP, and GFP-parkin was immunoprecipitated as in (**C**). Immunoprecipitates were then treated with Usp2 deubiquitinase and the beads were re-isolated by centrifugation. (**E**) Immunoblot detection of Mfn2 after on-bead deubiquitination, as described in (**D**). Immunoprecipitates were either incubated at 37°C in the absence or presence of Usp2 catalytic domain for 30 min. Samples were then centrifuged to separate beads and supernatant ('sup.'), which were denatured in sample buffer prior to separation by SDS-PAGE. Arrowheads indicate unmodified forms of Mfn2, while the red asterisks denote

*Figure 2 continued on next page*

*Figure 2 continued*

ubiquitinated forms. (**F**) Immunoprecipitation of Mfn2 for LC/MS analysis. Immunoprecipitates were separated, along with 4% input, by SDS-PAGE and immunoblotted for Ub. (**G**) Extracted ion chromatogram for the pS65 Ub peptide (TLSDYNIQKEpSTLHLVLR, a.a. 55–72) from Mfn2 immunoprecipitates from DMSO- (blue line) and CCCP- (red line) treated U2OS:GFP-parkin WT cells, immunoprecipitated as in (**F**). The red arrow indicates the peak corresponding to the peptide. (**H**) Immunoprecipitation of Mfn2 under denaturing conditions. Cells were lysed in buffer containing 1% SDS (see Materials and methods). Immunoprecipitates were separated, along with 4% input, by SDS-PAGE and immunoblotted for Ub and pS65 Ub. (**I**) Crystal structure of parkin complexed with pUb (PDB ID 5N2W, *Kumar et al., 2017*). The A320 residue at the pUb/parkin interface is highlighted in red, with parkin coloured blue and ubiquitin in green. (**J**) GST-R0RBR pulldown of pUb from U2OS:GFP-parkin WT cells. Pulldowns were performed with WT or A320R GST-R0RBR, with no GST-R0RBR ('-') as a further negative control. Pulldowns were separated, along with 10% input, by SDS-PAGE and immunoblotted for the indicated protein. The asterisk represents a cross-reaction between the pS65 antibody and the GST-R0RBR module.
DOI: https://doi.org/10.7554/eLife.32866.005

The following figure supplement is available for figure 2:

**Figure supplement 1.** LC/MS of immunoprecipitated Mfn2.
DOI: https://doi.org/10.7554/eLife.32866.006

took advantage of delayed pathway kinetics of respiring cells by culturing cells in growth medium containing galactose as a carbon source (rather than glucose). This forces ATP generation through the electron transport chain and mitigates parkin-dependent mitophagy (*Lee et al., 2015*; *McCoy et al., 2014*); mitochondrial translocation of parkin, and the buildup of Ub, p62 and LC3 on mitochondria are all slowed in galactose-grown cells (*Figure 3—figure supplement 2*). Remarkably, we observed faster mitochondrial recruitment in siMfn2 (but not siMfn1) cells, under both bioenergetic conditions (*Figure 3C and D*). A significant difference was visible within one hour of CCCP treatment in glucose-cultured cells, and was exacerbated in their galactose-grown counterparts, owing to their slower kinetics in the control siRNA-transfected condition (*Figure 3E*). Strikingly, Mfn2 silencing increased recruitment in galactose-grown cells to levels seen in glucose-maintained cells transfected with control siRNA (*Figure 3E*). Silencing Mfn1 and Mfn2 simultaneously (*Figure 3—figure supplement 3A*) did not further enhance the kinetics of parkin recruitment beyond single, Mfn2-depleted cells (*Figure 3—figure supplement 3B–D*), implying that this phenotype was Mfn2-specific and unrelated to a loss of mitochondrial fusion.

We next determined whether, more generally, this increase in recruitment kinetics could be induced by disrupting mitochondria-ER contacts via other means than removing Mfn2. To test this, we silenced two other genes that have been shown to promote mitochondria-ER association; PACS2 and Stx17 (*Figure 3—figure supplement 3E*) (*Arasaki et al., 2015*; *Simmen et al., 2005*). Unlike Mfn2 knockdown, we did not observe mitochondrial fragmentation in either PACS2- or Stx17-silenced cells (*Figure 3—figure supplement 3F*). When we tested parkin recruitment in these cells, we saw that, similarly to Mfn2 knockdown, silencing of either PACS2 (siPACS2) or Stx17 (siStx17) increased the translocation of parkin to mitochondria (*Figure 3—figure supplement 3G and H*). Again, the increase was most pronounced in galactose-cultured cells that were treated with CCCP for one hour, where parkin was recruited to near-glucose levels in Mfn2-, PACS2- and Stx17-silenced cells despite remaining predominantly cytosolic in cells transfected with control siRNA at this time point (*Figure 3—figure supplement 3G and I*). Thus, disruption of mitochondrion-ER tethering increases the kinetics of parkin translocation to depolarized mitochondria.

We next directly tested the effect of Mfn2 depletion on mitochondrial turnover using quantitative, ratiometric measurements of mitochondrially-targeted mKeima (mtKeima), a protein that shifts its fluorescence excitation when acidified by the lysosome (*Katayama et al., 2011*). We transfected U2OS cells stably-expressing mtKeima (U2OS:mtKeima), grown on either glucose or galactose, with siRNA targeting Mfn1 or Mfn2, followed by wild-type (WT) GFP-parkin, using the ligase-dead C431S mutant as a negative control. Next, we treated these cells with CCCP (or DMSO) for four hours and then determined the ratio of acidified mtKeima per cell by FACS (see Materials and methods) as a quantitative indicator of mitophagy (*Katayama et al., 2011*; *Tang et al., 2017*). As expected, in the glycolytic, CCCP-treated condition, a higher proportion of control siRNA-transfected cells had an increased ratio of acidified mtKeima compared with DMSO-treated counterparts (as these cells were undergoing mitophagy), and this population shift was similarly replicated in siMfn1 cells (*Figure 3F and G*). However, in Mfn2-depleted cells, we observed a ~ 2 fold increase in the proportion of cells undergoing mitophagy (*Figure 3F and G*). In respiring conditions, we did not observe a shift at all in

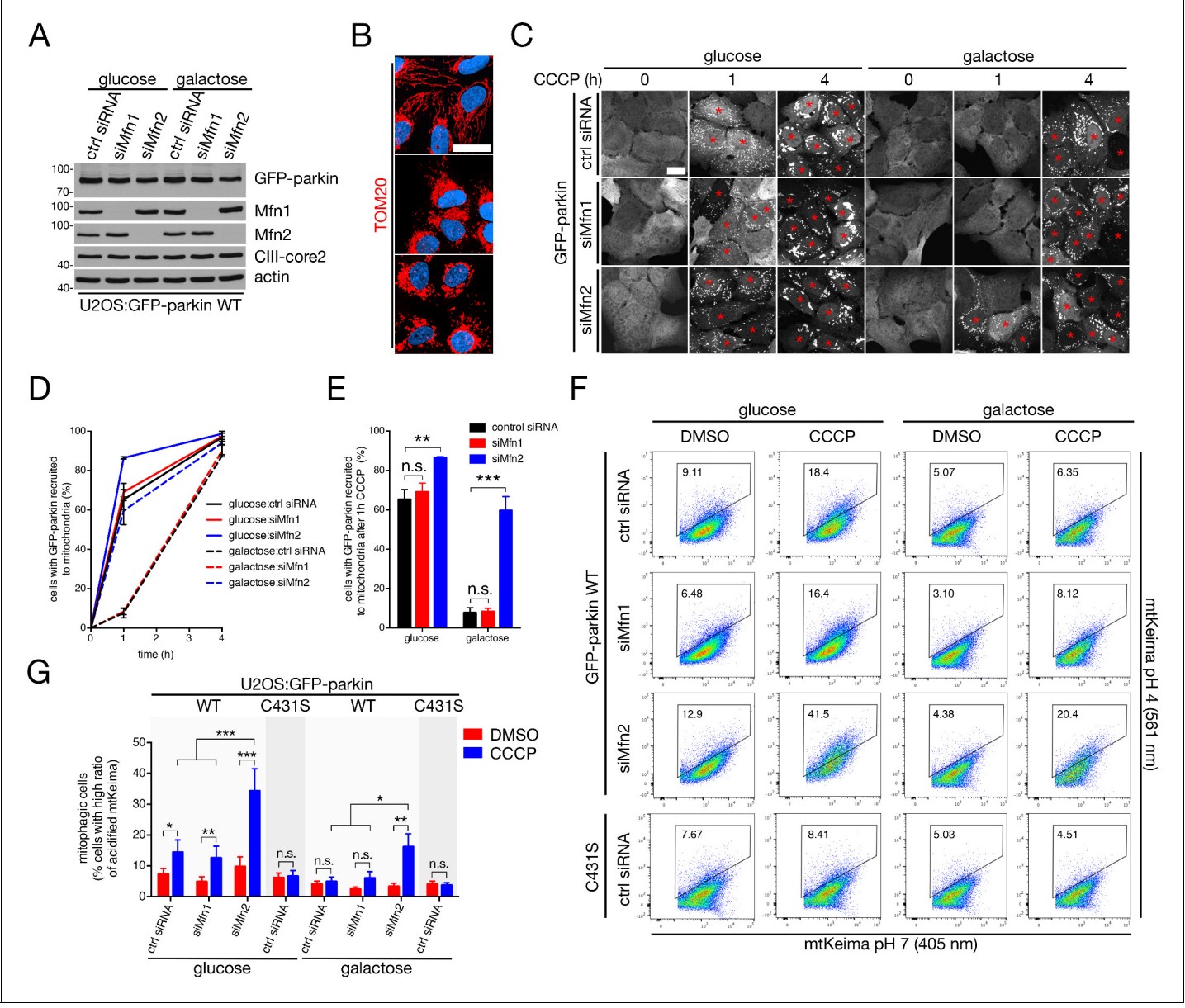

**Figure 3.** Mfn2 antagonizes mitophagy. (**A**) Immunoblot analysis of whole-cell lysates from cells cultured in glucose or galactose transfected with control siRNA or siRNA targeting Mfn1 ('siMfn1') or Mfn2 ('siMfn2'). (**B**) Mitochondrial morphology in glucose-maintained cells transfected with the indicated siRNA, as revealed by confocal imaging of TOM20 (red) staining (Hoechst, blue). Scale bar, 30 microns. (**C**) Representative confocal images of GFP-parkin recruitment to mitochondria as a function of time in U2OS:GFP-parkin cells treated with 20 μM CCCP. Red asterisks indicate cells in which GFP-parkin has fully translocated to mitochondria. Scale bar, 20 microns. (**D**) Quantification of parkin recruitment in cells from (**C**). Data points represent mean ± SEM, n = 3 replicates cells per condition, with >100 cells counted per condition for each replicate. (**E**) Parkin recruitment at one hour CCCP in cells from (**C**) arranged as a histogram. Bars represent mean ± SEM. n.s., not significant; **, p<0.01; ***, p<0.001. (**F**) U2OS:mtKeima cells were transfected with the indicated siRNA and GFP-parkin WT or C431S, and were treated with 20 μM CCCP (or DMSO) for four hours. mtKeima fluorescence in GFP-positive cells was measured using flow cytometry by excitation at 405 nm (neutral pH) and 561 nm (acidified). The data are represented as scatter plots of fluorescence emission from excitation at both wavelengths. The gated area encloses cells undergoing mitophagy (high acidified:neutral Keima ratio), and the percentage of cells within this gate is indicated in the top-left corner of each plot. (**G**) Quantification of the percent of cells undergoing mitophagy in cells from (**F**) treated with DMSO (red bars) or CCCP (blue bars) for four hours. Bars represent mean ± SEM, n = 2 experiments. n.s., not significant; *, p<0.05; **, p<0.01; ***, p<0.001.

DOI: https://doi.org/10.7554/eLife.32866.007

The following source data and figure supplements are available for figure 3:

**Source data 1.** Numerical source data for *Figure 3D, E and G*, *Figure 3—figure supplement 1B–D*, *Figures 2B*, *3C, D, H, I*, *4F, H* and *5C*.
DOI: https://doi.org/10.7554/eLife.32866.013

*Figure 3 continued on next page*

*Figure 3 continued*

**Figure supplement 1.** Mfn2 is a mitochondrion-ER tether.
DOI: https://doi.org/10.7554/eLife.32866.008
**Figure supplement 2.** Mitochondrial respiration impedes mitophagy.
DOI: https://doi.org/10.7554/eLife.32866.009
**Figure supplement 3.** Parkin recruitment kinetics in cells lacking both Mfns and other mitochondria-ER tethering factors.
DOI: https://doi.org/10.7554/eLife.32866.010
**Figure supplement 4.** Analysis of mitophagy in Mfn2 KO U2OS cells.
DOI: https://doi.org/10.7554/eLife.32866.011
**Figure supplement 5.** Parkin recruitment in Mfn2-depleted cells requires PINK1 and phosphoubiquitin binding.
DOI: https://doi.org/10.7554/eLife.32866.012

either control siRNA-transfected or Mfn1-depleted cells but observed a level of mitophagy in siMfn2 cells similar to control cells cultured in glucose medium (*Figure 3F and G*). These data demonstrate that, in Mfn2-depleted cells, depolarization-induced mitophagy is enhanced, in line with our parkin recruitment experiments (*Figure 3A to E*), and demonstrate that Mfn2 represses mitophagy at the level of pathway initiation.

To ensure that we were observing on-target effects from depletion of our siRNA targets, we replicated our recruitment data in Mfn2 knock-out (KO) U2OS cells that were generated using the CRISPR-Cas9 system (see *Materials and Methods*). Genetic disruption was confirmed by sequencing in two clones (A4 and A5) in which a premature stop codon was introduced via a single base-pair frame shift following the codon corresponding to leucine-29 in the human Mfn2 gene (*Figure 3—figure supplement 4A*). We validated these KO cells by immunoblot, along with a clone that underwent the complete procedure and selection but in which Mfn2 knock out failed (B4) as a further negative control; importantly, Mfn1 levels remained similar across all lines, and the core subunits of the mitochondrial $Ca^{2+}$ uniporter remained unperturbed (*Figure 3—figure supplement 4B*, compensation in the latter has been reported in MEFs isolated from Mfn2$^{-/-}$ mice [*Filadi et al., 2015*]). Accordingly, Mfn2 KO cells had mitochondrial reticula that were similarly polarized but fragmented compared to WT U2OS cells (*Figure 3—figure supplement 4C and D*). Corroborating our earlier data in siMfn2 cells, Mfn2 KO cells (grown on glucose) transiently transfected with GFP-parkin displayed increased recruitment kinetics (*Figure 3—figure supplement 4E and F*) and increased mitophagy (*Figure 3—figure supplement 4G and H*). Finally, we ensured that parkin translocation in Mfn2 KO cells (*Figure 3—figure supplement 5A–C*) and U2OS:GFP-parkin cells depleted of Mfn2 (*Figure 3—figure supplement 5D*) remained PINK1-dependent. Moreover, cells expressing GFP-parkin$^{A320R}$ (*Figure 3—figure supplement 5E*) failed to translocate under conditions of Mfn2-depletion (*Figure 3—figure supplement 5F and G*). This indicates a clear requirement for PINK1 and Ub phosphorylation for parkin translocation in Mfn2-depleted cells, demonstrating that Mfn2 reduction increases on-pathway mitophagy kinetics. Taken together, our data not only show that mitochondria-ER contact is dispensable for mitophagy, but that this type of organellar coupling in fact antagonizes the pathway.

We next sought to demonstrate that the antagonistic effect of mitochondria-ER tethering on mitophagy was functioning directly through the degradation of Mfn2. Conceivably, we could manipulate the pathway by preventing ER-OMM dissociation through the blockage of Mfn2 turnover, which is mediated by proteasomal degradation coupled to parkin ubiquitination (*Tanaka et al., 2010*; *Ziviani et al., 2010*). This hypothesis is supported by our EM data demonstrating that MG132 blocks mitochondria-ER uncoupling during mitophagy (*Figure 1H–J*). To achieve this, we created Mfn2 KO cells stably-expressing YFP-parkin (Mfn2 KO:YFP-parkin) and re-expressed ectopic Mfn2, which was able to rescue mitochondrial morphology from a fragmented reticulum to a collection of tubules (*Figure 4A*; CFP is used to identify cells expressing untagged Mfn2). We could additionally rescue morphology by overexpression of Mfn1 (*Figure 4A*), a phenomenon that has been described previously (*Chen et al., 2003*). Turning to recruitment assays – in which we observed faster GFP-parkin recruitment in Mfn2 KO cells (*Figure 3—figure supplement 4E and F*) – we observed that ectopic expression of Mfn2, but not Mfn1, was able to suppress the recruitment of YFP-parkin to depolarized mitochondria (*Figure 4B and C*). This is in line with our previous data showing that the antagonistic effect of Mfn2 on mitophagy occurs through its ability to tether mitochondria to the ER

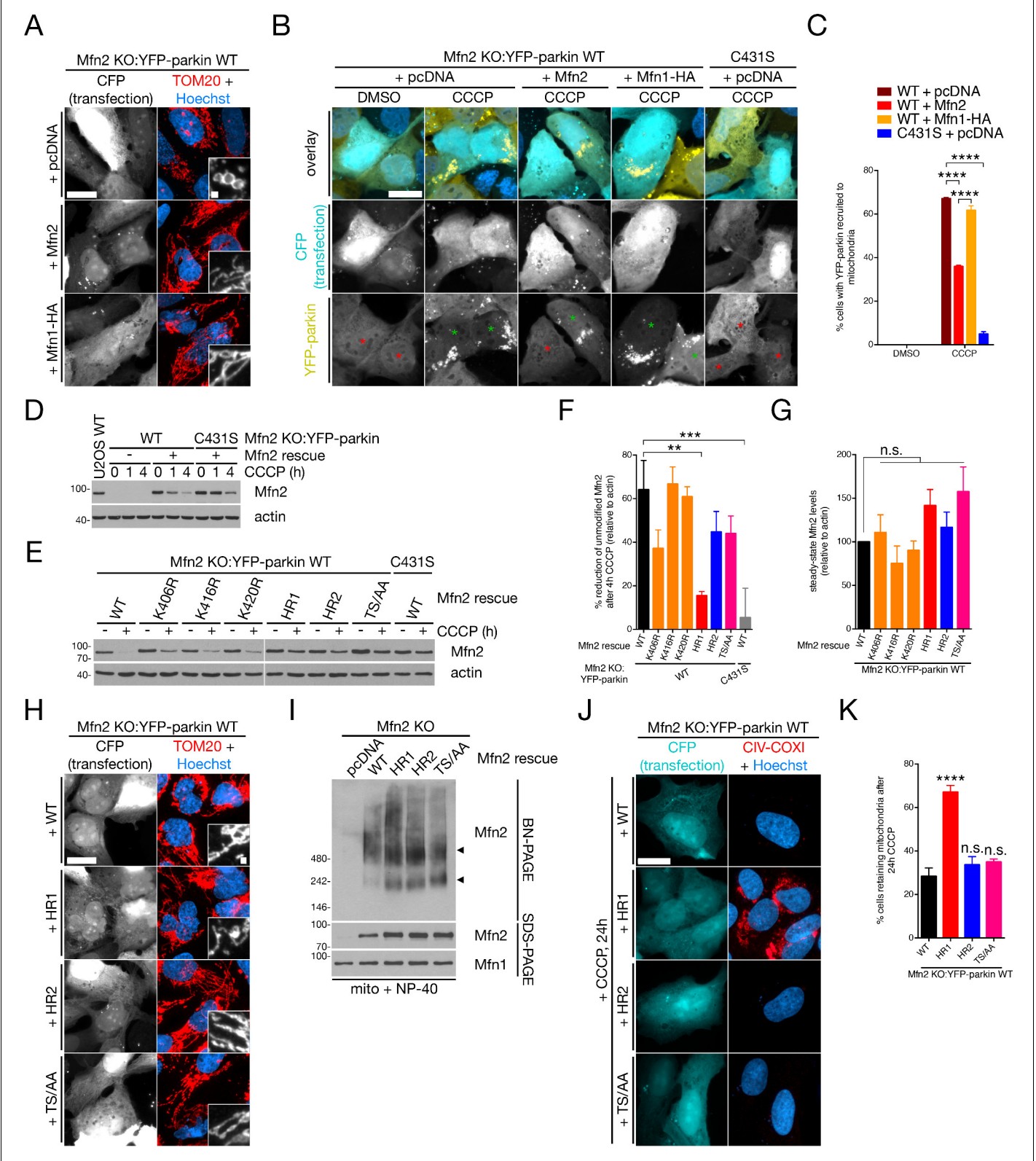

**Figure 4.** Parkin ubiquitinates Mfn2 in the HR1 domain to derepress mitophagy. (**A**) Mfn2 KO:YFP-parkin WT cells were transfected with the indicated plasmid and CFP in a 3:1 ratio, then fixed and immunostained for TOM20 (red) and counterstained with Hoechst 33342 (blue). Scale bars, 20 and 1 microns. (**B**) Mfn2 KO:YFP-parkin WT and C431S cells, transfected as in (**A**), were treated with 20 μM CCCP for four hours prior to fixation, then scored for YFP-parkin recruitment. Green and red asterisks indicated CFP-positive cells with mitochondrial and cytosolic YFP-parkin, respectively. Scale bar, 20

*Figure 4 continued*

microns. (C) Quantification of recruitment in (B). Bars represent mean ± SEM, n = 3 replicates cells per condition, with >50 cells counted per condition for each replicate. ****, p<0.0001. (D) Immunoblot analysis of Mfn2 KO:YFP-parkin cells (WT and C431S) transfected with Mfn2 and treated with 20 μM CCCP for the indicated time. An untreated U2OS cell lysate is included as a control for endogenous Mfn2 levels. (E) Representative immunoblot analysis of Mfn2 KO:YFP-parkin WT cells transfected with the indicated Mfn2 mutant and treated with 20 μM CCCP for four hours. (F) Quantification of Mfn2 modification in immunoblot analyses from (E), given as the percent of Mfn2 reduction after CCCP relative to actin. Bars represent mean ± SEM, n = 4 replicates. **, p<0.01; ***, p<0.001. (G) Quantification of steady-state ('- CCCP') levels of Mfn2 in immunoblot analyses from (E), relative to actin. Bars represent mean ± SEM, n = 4 replicates. n.s., not significant. (H) Mfn2 KO:YFP-parkin WT cells were transfected with the indicated plasmid and CFP in a 3:1 ratio, then fixed and immunostained for TOM20 (red) and counterstained with Hoechst 33342 (blue). Scale bars, 20 and 1 microns. (I) Immunoblot analysis of BN- and SDS-PAGE gels of solubilized mitochondria from cells from (H). Arrows indicate two Mfn2-containing complexes in the native condition. (J) Representative wide-field images of Mfn2 KO:YFP-parkin WT cells transfected with the indicated Mfn2 construct. Cells were treated with 20 μM CCCP for 24 hr prior to fixation, then stained with CIV-COX1 (red) and Hoechst (blue). Scale bar, 20 microns. (K) Quantification of mitophagy in (J). Bars represent mean ± SEM, n = 4 replicates per condition, with >50 cells counted per condition for each replicate. ****, p<0.0001; n. s., not significant.

DOI: https://doi.org/10.7554/eLife.32866.014

The following source data and figure supplement are available for figure 4:

**Source data 1.** Numerical source data for *Figure 4C, F, G and K*.
DOI: https://doi.org/10.7554/eLife.32866.016

**Figure supplement 1.** Location and conservation of ubiquitination and phosphorylation sites in Mfn2.
DOI: https://doi.org/10.7554/eLife.32866.015

(*Figure 3—figure supplement 3H–J*) and not its effect on mitochondrial fusion (*Figure 3—figure supplement 3A–D*). Immunoblot analysis of Mfn2 KO:YFP-parkin[WT] cells ectopically expressing Mfn2 revealed that it was expressed at near-endogenous levels and degraded rapidly upon CCCP treatment compared to the control Mfn2 KO:YFP-parkin[C431S] cell line (*Figure 4D*). Mfn2 is ubiquitinated by parkin on at least ten lysine residues, although several sites are clustered in the heptad repeat (HR) domains (*Sarraf et al., 2013*). Additionally, Mfn2 itself has been reported to be directly phosphorylated by PINK1 on T111 and S442, and that these phosphorylation events are critical for the interaction of parkin with Mfn2 and parkin recruitment in cardiomyocytes (*Chen and Dorn, 2013*). Focusing on these putative phosphorylation sites and the clustered ubiquitination sites in the HR1 and HR2 domains, phylogenic analysis of their conservation demonstrated that only T111 in the GTPase domain and K737 in the HR2 domain were completely conserved from human Mfn2 to the sole *Drosophila* mitofusin, MARF (*Figure 4—figure supplement 1A and B*; both the traditional and single-pass Mfn2 topologies (*Mattie et al., 2018*) are depicted in *Figure 4—figure supplement 1B*). However, in the case of the sites of ubiquitination, at least two HR1 sites and three HR2 sites were conserved as lysines down through *Xenopus* Mfn2, while MARF retained one site each in HR1 and HR2 (*Figure 4—figure supplement 1A*). We thus posited that mutation of several lysine residues would likely be required to reduce Mfn2 ubiquitination. While mutation of all major sites of Mfn2 ubiquitination almost completely abolishes its modification by parkin (*Heo et al., 2015*), we found that mutation of K406, K416 and K420 in the HR1 domain (Mfn2[HR1]) reduced its CCCP-induced ubiquitination by ~75%, as measured by the disappearance of the unmodified band by immunoblot (*Figure 4E and F*; here Mfn2 levels are normalized to the untreated condition for each construct). This effect was greater than what we observed with the single mutant, Mfn2[K406R] (K416 and K420 appear dispensable in this assay), and mutation of all four sites in HR2 (Mfn2[HR2]) or the double T111A/S442A phosphomutant (Mfn2[TS/AA]) failed to significantly reduce Mfn2 modification (*Figure 4E and F*). We thus considered Mfn2[HR1] as a 'hypomorph' with respect to parkin ubiquitination compared to WT, HR2 and TS/AA constructs, despite similar expression patterns with the latter two (*Figure 4G*). Introduction of either Mfn2[HR1], Mfn2[HR2] or Mfn2[T111A/S442A] into Mfn2 KO:YFP-parkin cells rescued morphology in a similar manner to WT Mfn2 (*Figure 4H*), demonstrating these mutations did not disrupt mitochondrial fusion. We also monitored the ability of these Mfn2 mutants to form high molecular weight (HMW) complexes (*Karbowski et al., 2006*) that function in mitochondria-ER tethering (*de Brito and Scorrano, 2008*). By blue native polyacrylamide gel electrophoresis (BN-PAGE), we observed that all three mutants (HR1, HR2 and T111A/S442A) formed HMW complexes similar to WT in solubilized mitochondria (*Figure 4I*). When we assayed mitophagy in Mfn2 KO:YFP-parkin[WT] cells, we found that only rescue of Mfn2 with Mfn2[HR1] – the ubiquitination of

which is compromised (*Figure 4E and F*) – blocked the turnover of mitochondria (*Figure 4J and K*). Thus, ubiquitination of the Mfn2 HR1 domain by parkin is required for efficient mitophagy and, taken together with our previous mitophagic data in Mfn2-depleted cells, demonstrates that parkin and PINK1 directly counter Mfn2-mediated mitochondria-ER tethering through Mfn2 turnover to promote mitophagy.

## Mfn2 complexes are extracted by p97 to drive mitochondria and the ER apart

We next investigated exactly how parkin and PINK1 act on Mfn2-mediated OMM-ER tethering. Examining HMW complexes by BN-PAGE in untreated U2OS:GFP-parkin^WT cells (expressing endogenous Mfn2), we observed a bimodal distribution of Mfn2 into two complexes, weighing approximately ~250 kDa and ~500 kDa (*Figure 5A*, leftmost lane, similar to what was seen in *Figure 4I*). By contrast, Mfn1 – which, in our assays, appears dispensable for mitochondria-ER tethering as assayed by EM (*Figure 3—figure supplement 1*) and its effect on parkin recruitment (*Figure 3C–E*) – only formed a ~ 250 kDa HMW complex (*Figure 5A*). We thus considered the ~500 kDa complex containing solely Mfn2 as a dimer of the ~250 kDa Mfn2-containing subcomplex that potentially bridges the ER and OMM. We then monitored the stability of Mfn2- (and Mfn1-) containing HMW complexes during mitophagy. Upon CCCP treatment, we observed a rapid loss Mfn2- (and Mfn1-) containing complexes (*Figure 5A*), concomitant with its phosphoubiquitination (*Figure 2*) and dependent upon parkin ligase activity (*Figure 5B and C*). While treatment of mitochondrial lysates with Usp2 deubiquitinase slightly increased levels of the unmodified Mfn1 or Mfn2 band in mitochondria isolated from CCCP-treated cells (*Figure 5D*; the densitometry measurements correspond to the shorter exposures of Mfn1 and Mfn2), this was not to levels seen in mitochondria from untreated cells. This result indicated that the disappearance of HMW Mfn complexes are predominantly due to their extraction from the OMM (and not a high level of modification by Ub). This process is thought to be mediated by the AAA-ATPase p97/VCP (*Tanaka et al., 2010*) and, accordingly, when we treated U2OS:GFP-parkin^WT cells with CCCP in the presence of the non-competitive p97 inhibitor NMS-873 (*Magnaghi et al., 2013*), extraction of HMW complexes containing either Mfn1 or Mfn2 was accordingly repressed (*Figure 5E*). Indeed, both ~250 kDa (containing Mfn1 and/or Mfn2) and ~500 kDa (Mfn2 only) complexes were stabilized in the presence of NMS-873 (*Figure 5E*), with smearing occurring due to Mfn ubiquitination (see *Figure 2*), indicating that parkin-mediated ubiquitination itself was not sufficient to drive apart the ~500 kDa Mfn2-containing interorganellar bridge. Analysis of OMM-ER appositions in these cells revealed that p97 inhibition prevented the dissociation of mitochondria from the ER (*Figure 5F–H*). Thus, p97-dependent extraction of Mfn2 HMW complexes from the OMM separates mitochondria from the ER during mitophagy.

We then addressed the relationship between parkin-dependent Mfn2 ubiquitination and p97 extraction more closely. Consistent with our HMW complex extraction data (*Figure 5E*), co-incubation of cells with CCCP and NMS-873 completely blocked the mitochondrial translocation of p97 (*Figure 6A and B*) which occurs during mitophagy (*Kimura et al., 2013*; *Tanaka et al., 2010*). Accordingly, NMS-873 stabilized ubiquitinated Mfn1 and Mfn2 conjugates induced by CCCP in whole-cell extracts (*Figure 6C*) and, consistent with our BN-PAGE data (*Figure 5E*), these ubiquitinated Mfn2 species were present on mitochondria (*Figure 6D*). We observed a similar effect when we silenced p97 with siRNA (sip97); in p97-depleted cells treated with CCCP, we saw an increase in ubiquitinated Mfn2 upon depolarization (*Figure 6E*). Additionally, basal levels of Mfn2 increased upon prolonged p97 depletion (*Figure 6E*), consistent with the possible involvement of p97 in steady-state Mfn2 turnover (*Zhang et al., 2017*). In Mfn2 KO:YFP-parkin^WT cells rescued with WT Mfn2, CCCP induced Mfn2 turnover and, when cells were co-incubated with NMS-873, we observed a stabilization of ubiquitinated Mfn2 (*Figure 6F*) similar to WT U2OS cells expressing GFP-parkin (*Figure 6C*). When we expressed Mfn2^HR1 in Mfn2 KO:YFP-parkin^WT cells, we observed a severe reduction in NMS-873-dependent stabilization of CCCP-induced Mfn2-Ub conjugates (*Figure 6F*). We confirmed this reduction in ubiquitination by immunoprecipiting Mfn2 from reconstituted cells treated with CCCP and NMS-873 under denaturing conditions and immunoblotting for Ub (*Figure 6G*). This supports our mutagenesis data showing a reduction of Mfn2^HR1 turnover (*Figure 4E and F*) and is mechanistically consistent with ubiquitination of lysines in the Mfn2 HR1 domain being recognized by p97 and signaling for extraction of the protein.

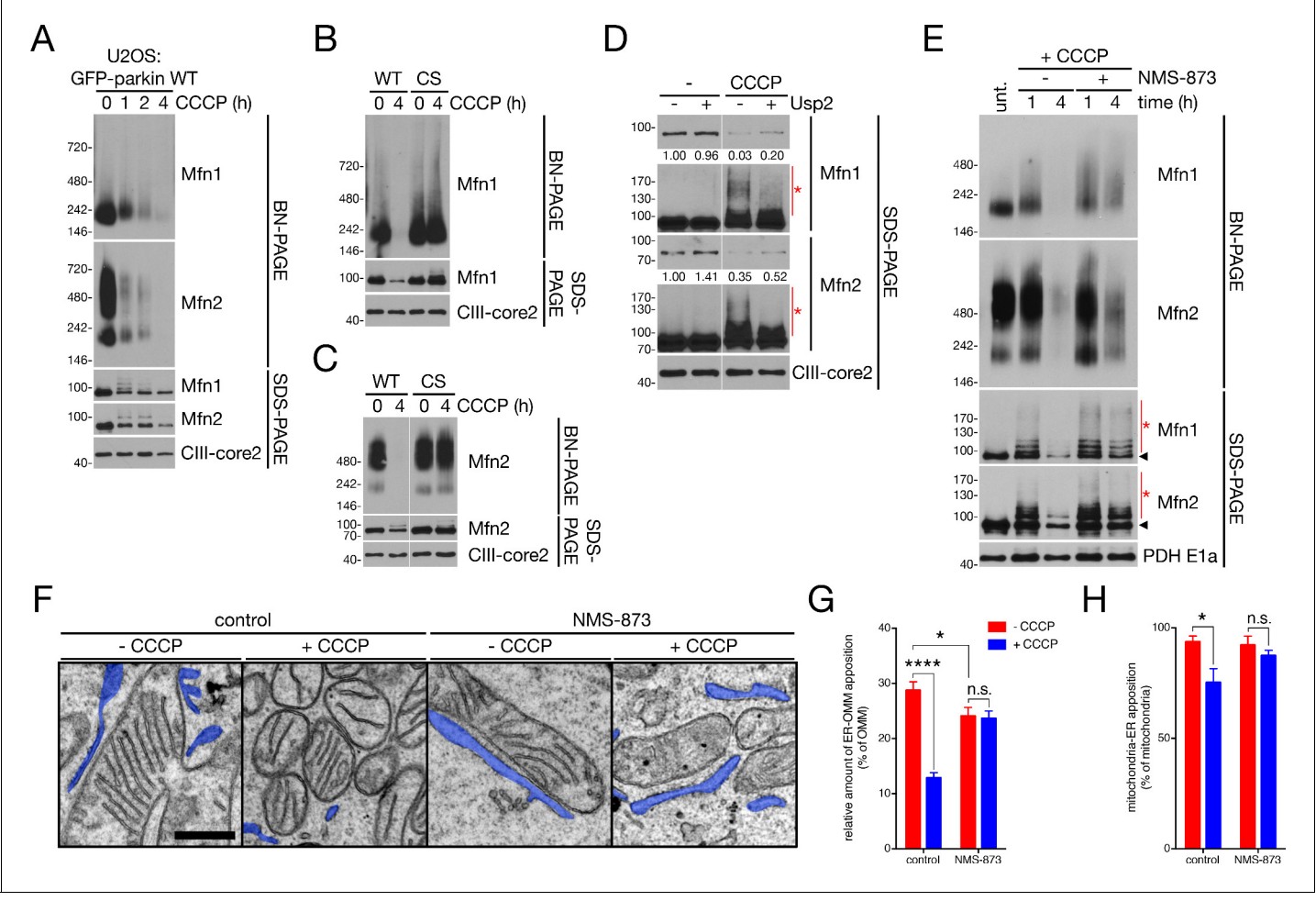

**Figure 5.** p97 governs ER-OMM contact via the extraction of Mfn2 complexes. (**A**) Immunoblot analysis of NP-40-solubilized mitochondria, isolated from U2OS:GFP-parkin WT cells treated with 20 μM CCCP for the indicated time, separated by blue native- (BN-) and SDS-PAGE. (**B, C**) Immunoblot analysis of Mfn1- (**B**) and Mfn2- (**C**) containing complexes in NP-40-solubilized mitochondria, isolated from U2OS:GFP-parkin WT and C431S cells treated with 20 μM CCCP for four hours, separated by BN- and SDS-PAGE. (**D**) Mitochondria isolated from U2OS:GFP-parkin WT cells treated with 20 μM CCCP for one hour were, after solubilization in NP-40, incubated with 1 μM Usp2 for 30 min at 37°C prior to separation by SDS-PAGE. Red asterisks indicate ubiquitinated species of Mfn1 and Mfn2. Densitometry calculations for the Mfn1 and Mfn2 bands (shorter exposure) relative to CIII-core2 are shown under the respective immunoblots. (**E**) Immunoblot analysis of NP-40-solubilized mitochondria, isolated from U2OS:GFP-parkin WT cells treated with 20 μM CCCP in the presence or absence of 25 μM NMS-873 for the indicated time, separated by blue native- (BN-) and SDS-PAGE. Red asterisks indicate ubiquinated Mfn species visible by SDS-PAGE, while the arrowhead denotes the unmodified band. (**F**) Representative TEM images of mitochondria in contact with ER (pseudocoloured blue) in U2OS:GFP-parkin cells treated with 20 μM CCCP ('+CCCP') for four hours in the presence or absence of 25 μM NMS-873. Scale bar, 500 nm. (**G,H**) Quantification of TEM from (**F**) in cells treated with (blue bars) or without (red bars) 20 μM CCCP for four hours. The percent of OMM per mitochondrion (**G**) and mitochondria per field (**H**) in contact with the ER was quantified. Bars represent mean ± SEM, n = 99 to 187 mitochondria in 12 to 14 fields per condition. n.s., not significant; *, p<0.05; ***, p<0.001; ****, p<0.0001.
DOI: https://doi.org/10.7554/eLife.32866.017

The following source data is available for figure 5:

**Source data 1.** Numerical source data for *Figure 5G and H*.
DOI: https://doi.org/10.7554/eLife.32866.018

While we posited that Mfn2 may be acting as a p97 receptor during mitophagy, we observed robust p97 recruitment in depolarized Mfn2 KO:YFP-parkin[WT] cells (*Figure 6H and I*). Moreover, p97 recruitment was similar in cells expressing either Mfn2[WT] or Mfn2[HR1] (*Figure 6H and I*). p97 recruitment levels in both Mfn2 rescue conditions were lower than in cells transfected with empty vector (*Figure 6H and I*) likely owing to the delayed parkin recruitment kinetics in Mfn2-expressing cells (*Figure 4B and C*). Thus, ubiquitinated Mfn2 is not the sole p97-binding protein on the OMM.

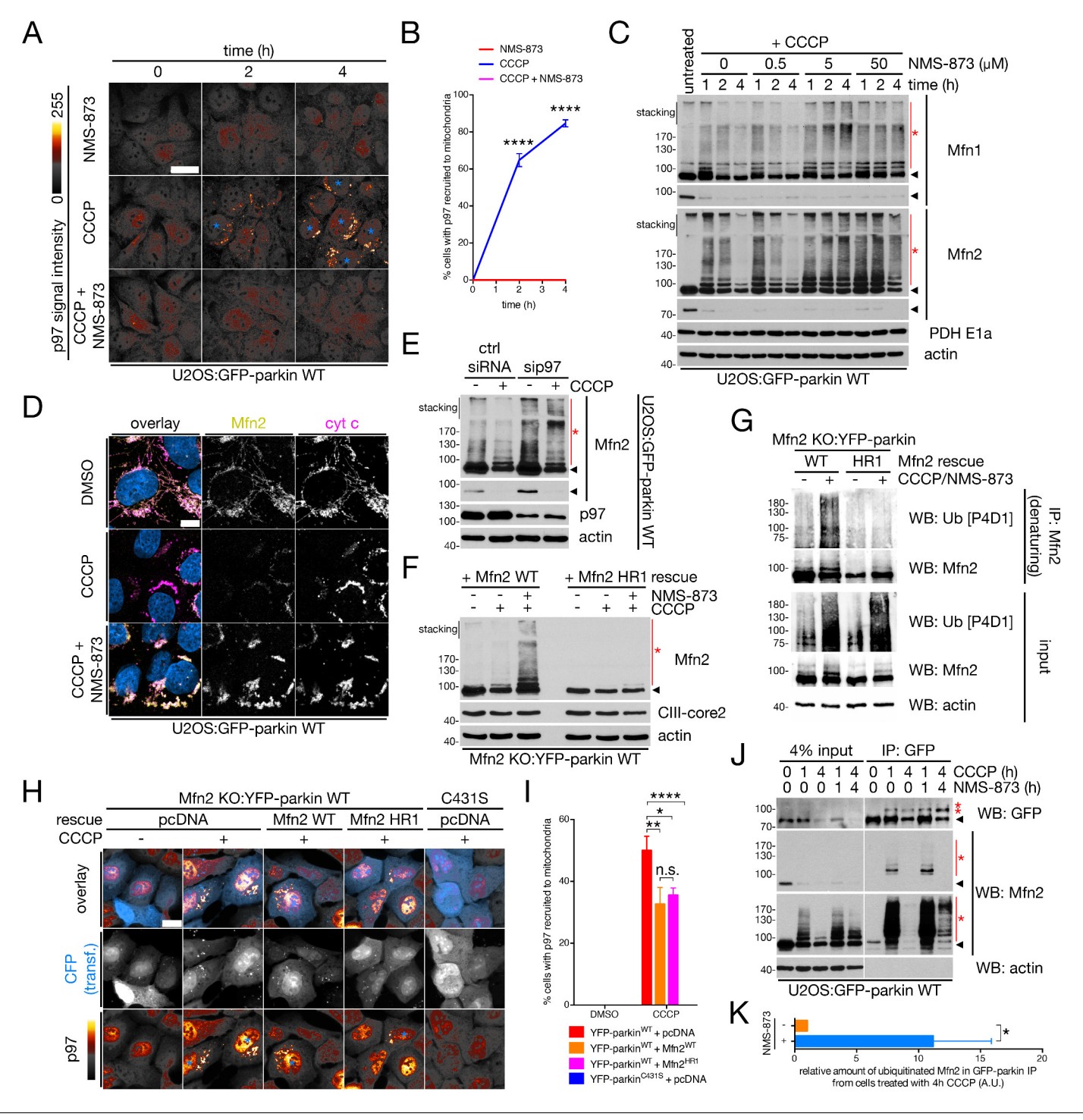

**Figure 6.** Degradation of ubiquitinated Mfn2 involves p97 recruitment and activity. (**A**) Representative confocal images of p97 recruitment to mitochondria in cells treated with 20 μM CCCP and/or 25 μM NMS-873 for the indicated time. Blue asterisks denote cells with mitochondrial p97, and p97 signal intensity is represented as a heat map. Scale bar, 20 microns. (**B**) Quantification of cells with p97 translocation to mitochondria in cells treated with either 25 μM NMS-873 (red line), 20 μM CCCP (blue line) or both simultaneously (magenta line). Bars represent mean ± SEM, n = 3 replicates per condition, with >100 cells counted per condition for each replicate. ****, p<0.0001. (**C**) Immunoblot analysis of whole-cell lysates from U2OS:GFP-parkin cells treated with 20 μM CCCP and the specified concentration of NMS-873 for the indicated time, separated by SDS-PAGE. For each Mfn, longer (upper panel) and shorter (lower panel) exposures are shown. Red asterisks indicate ubiquitinated Mfn species, while the arrowheads denote the unmodified band. (**D**) U2OS:GFP-parkin cells were treated with 20 μM CCCP in the presence or absence of 25 μM NMS-873 for four hours, then fixed and immunostained for Mfn2 (yellow) and cytochrome c (magenta). Scale bar, 10 microns. (**E**) Immunoblot analysis of Mfn2 ubiquitination in

*Figure 6 continued on next page*

*Figure 6 continued*

U2OS:GFP-parkin WT cells transfected with siRNA targeting p97 (sip97) or control (ctrl siRNA) and treated with 20 µM CCCP for two hours. Arrowheads indicate the unmodified Mfn2 band (two exposures), while the red asterisk denotes ubiquitinated Mfn2. (F) Immunoblot analysis of exogenous Mfn2 in Mfn2 KO:YFP-parkin WT cells reconstituted with the indicated Mfn2 construct. Cells were treated with 25 µM NMS-873 and/or 20 µM CCCP for four hours prior to lysis. The arrowhead indicates the unmodified Mfn2 band and the red asterisk denotes ubiquitinated Mfn2 conjugates. (G) Immunoprecipitation of Mfn2 under denaturing conditions from Mfn2 KO:YFP-parkin WT cells reconstituted with the indicated Mfn2 construct. Cells were lysed in buffer containing 1% SDS (see *Materials and Methods*). Immunoprecipitates were separated by SDS-PAGE and immunoblotted for Ub. (H) Representative widefield images of p97 translocation to mitochondria (pseudocoloured as in [A]) in Mfn2 KO:YFP-parkin WT or C431S cells, reconstituted with the indicated plasmid, treated with 20 µM CCCP (or DMSO) for four hours. CFP (blue) is included as a marker of Mfn2 transfection, and blue asterisks indicate cells where p97 has translocated to mitochondria. Scale bar, 20 microns. (I) Quantification of mitochondrial recruitment of p97 in Mfn2 KO:YFP-parkin cells from (H). Bars represent mean ± SEM, n = 3 replicates per condition, with >50 cells counted per condition for each replicate. *, p<0.05; **, p<0.01; ****, p<0.0001. (J) Co-immunoprecipitation of mitofusins with GFP-parkin in U2OS:GFP-parkin WT cells treated with 20 µM CCCP in the presence or absence of 25 µM NMS-873 for the indicated time, using an anti-GFP antibody. Immunoprecipitates were separated, along with 4% input, by SDS-PAGE and immunoblotted for the indicated protein. The arrowhead indicates the unmodified form of the protein, while the asterisks denote ubiquitinated forms. (K) Quantification of the relative amount of ubiquitinated Mfn2 co-immunoprecipitated with GFP-parkin in cells from (J). Bars represent mean ± SEM, n = 3 replicates. *, p<0.05.

DOI: https://doi.org/10.7554/eLife.32866.019

The following source data and figure supplement are available for figure 6:

**Source data 1.** Numerical source data for *Figure 6I and K* and *Figure 6—figure supplement 1C*.
DOI: https://doi.org/10.7554/eLife.32866.021

**Figure supplement 1.** Analysis of pUb interactors from mouse brain.
DOI: https://doi.org/10.7554/eLife.32866.020

We next tested if pUb moieties conjugated to Mfn2 play a role in p97 binding. As we detected pUb conjugated to immunoprecipitated Mfn2 from cells treated with CCCP (*Figure 2G and H*), we co-treated cells with CCCP and NMS-873 and observed that the interaction between parkin and ubiquitinated Mfn2 – which is normally transient owing to Mfn2 turnover – was stabilized (*Figure 6J and K*). Finally, we probed for the existence of a pUb-p97 interaction by performing a GST pull-down using either S65-phosphorylated or unphosphorylated 4xUb chains from mouse brain lysate (see *Figure 6—figure supplement 1A* for experimental schematic) and identified interactors by LC/MS. Using nearly fully-phosphorylated chains (*Figure 6—figure supplement 1B*), we consistently observed the presence of p97, as well as its cofactors p47 and UBXN1, in 4xUb pull-downs, and these proteins were almost totally absent in parallel 4xpUb pull-downs (*Figure 6—figure supplement 1C* and *Supplementary file 1*). Thus, while p97 mediates the turnover of ubiquitinated Mfn2, this likely does not involve interactions between the p97 complex and pUb.

The herein-described role of p97 in separating mitochondria from the ER is critical; parkin-mediated ubiquitination on its own appears to be insufficient to drive the disassembly of Mfn2 HMW complexes (*Figure 5E*) or to dissociate the ER from the OMM (*Figure 5F and G*) in the absence of p97 activity. To clarify the role of p97 in mitophagy, we investigated the potentially epistatic relationship between p97 and Mfn2. We first measured mitophagy in U2OS:mtKeima cells expressing GFP-parkin[WT], comparing the effect of p97 inhibition in cells depleted of Mfn2 to control cells. In control siRNA-transfected cells, inhibition of p97 by NMS-873 abolished the CCCP-dependent,~3 fold increase in cells with acidified mtKeima (*Figure 7A and B*, red and orange bars in *Figure 7B*). When cells were depleted of Mfn2 (siMfn2), p97 inhibition reduced the rate of mtKeima acidification (*Figure 7A and B*, dark and light blue bars), but mitophagy was still permissive. Indeed, the number of cells with acidified mtKeima in siMfn2 cells treated with NMS-873 was still ~5 fold greater than their DMSO treated counterparts (*Figure 7B*, light blue bar), which was more of an increase that was observed for control cells with active p97 (*Figure 7B*, red bar). Thus, in the absence of Mfn2, inhibition of p97 fails to suppress mitophagy, demonstrating that a significant component of the role of p97 in mitophagy functions through Mfn2. As p97 extracts Mfn2-containing interorganellar bridges to uncouple mitochondria from the ER (*Figure 5*), we reasoned that Mfn2-mediated mitochondria-ER tethering may restrict the parkin-mediated ubiquitination of specific OMM substrates. Thus, we analyzed a sample of parkin substrates by immunoblot in CCCP-treated cells depleted of Mfn2 compared to control, in the presence or absence of NMS-873 (*Figure 7C*). We observed that the parkin-dependent ubiquitination of VDAC1 – which has been reported to form a complex with pUb and parkin that is stable over a period of hours (*Callegari et al., 2017*) – was sensitive to p97 inhibition

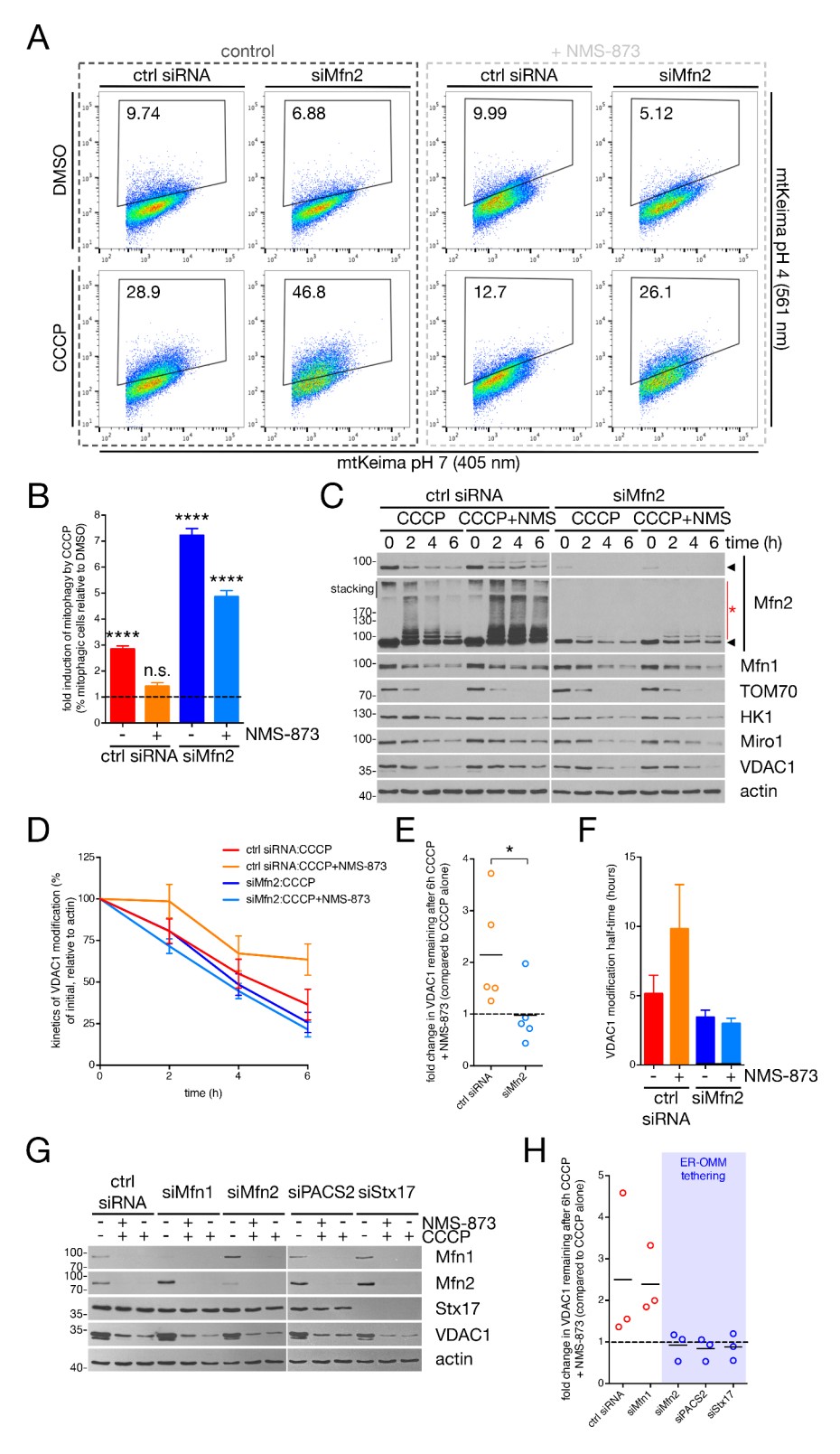

**Figure 7.** p97 and Mfn2 effect mitophagy through parkin substrate availability. (**A**) U2OS:mtKeima cells were transfected with the indicated siRNA and GFP-parkin WT, and were treated with 20 μM CCCP (or DMSO) for five hours in the presence (dark grey box) or absence (light grey box) of 25 μM NMS-873. mtKeima fluorescence in GFP-positive cells was measured using flow cytometry by excitation at 405 nm (neutral pH) and 561 nm (acidified). The data are represented as scatter plots of fluorescence emission from excitation at both wavelengths. The gated area encloses cells undergoing

*Figure 7 continued on next page*

**Figure 7 continued**

mitophagy and the percentage of cells within this gate is indicated in the top-left corner of each plot. (B) Quantification of the percent of cells undergoing mitophagy in cells from (A), expressed as a ratio of CCCP-treated cells to those treated with DMSO. Bars represent mean ± SEM, n = 2 experiments. n.s., not significant; ****, p<0.0001. (C) Immunoblot analysis of U2OS:GFP-parkin cells, transfected with siRNA targeting Mfn2 (siMfn2) or control (ctrl siRNA), treated with 20 µM CCCP in the presence or absence of 25 µM NMS-873 over a period of six hours. (D) Immunoblot quantification of VDAC1 levels (relative to actin) from cells from (C). Bars represent mean ± SEM, n = 5 experiments. (E) The 6 hr time-point data from (D) is represented as a fold change in VDAC1 remaining when NMS-873 is added. Data points are represented on the graph, n = 5 experiments. *, p<0.05. (F) Quantification of the half-time ($t_{1/2}$) of VDAC1 modification in cells from (C) over 6 hr. Half-times were obtained from decay curves generated with the time-points in (C). Bars represent mean ± SEM, n = 5 experiments. (G) Immunoblot analysis of U2OS:GFP-parkin cells, transfected with the indicated siRNA, treated with 20 µM CCCP in the presence or absence of 25 µM NMS-873 for six hours. (H) Immunoblot quantification of VDAC1 levels (relative to actin) in cells from (G), represented as a fold change in VDAC1 remaining when NMS-873 is added. Data points are represented on the graph, n = 3 experiments. Factors promoting ER-OMM contact are contained within the blue box.

DOI: https://doi.org/10.7554/eLife.32866.022

The following source data is available for figure 7:

**Source data 1.** Numerical source data for *Figure 7B, D–F and H*.

DOI: https://doi.org/10.7554/eLife.32866.023

in control cells, but not cells depleted of Mfn2 (*Figure 7C–E*). Indeed, the half-time of VDAC1 modification during mitophagy increased two-fold in the presence of NMS-873 specifically in control cells compared to cells transfected with siMfn2 (*Figure 7F*). We observed a similar effect pertaining to the difference in CCCP-dependent VDAC1 modification between cells treated with NMS-873 *versus* control across all cells depleted of promoters of mitochondria-ER tethering (Mfn2, PACS2 and Stx17) (*Figure 7G and H*). Notably, cells depleted of Mfn1 were comparable to control siRNA-transfected cells in this regard (*Figure 7G and H*). Thus, p97 relieves Mfn2-dependent inhibition of the ubiquitination of VDAC1 (and likely other OMM substrates). In this manner, Mfn2 gates the availability of the stable parkin receptor VDAC1 (*Callegari et al., 2017*), and mechanistically reconciles our data concerning the destruction of ER-OMM contacts during mitophagy, Mfn2-dependent mitophagy inhibition, and p97-mediated facilitation of ER-OMM uncoupling.

## Cell-free reconstitution of Mfn2 and VDAC1 ubiquitination by PINK1/parkin/p97

Cell-free reconstitution assays have proven useful in interrogating the activation of parkin-dependent ubiquitination by both PINK1 (*Lazarou et al., 2013*) and designer mutations in parkin itself (*Tang et al., 2017*). We thus sought to recapitulate our findings in cells concerning Mfn2 and VDAC1 ubiquitination in a cell-free assay (see diagram in *Figure 8A*). We first isolated mitochondria from HeLa cells – which lack endogenous parkin (*Denison et al., 2003*) – that were either depolarized with CCCP for four hours ('mito$^{CCCP}$') or treated with DMSO as a control ('mito$^{DMSO}$'). Accordingly, we observed PINK1 stabilization in the CCCP-treated condition only (*Figure 8B*). We were then able to reconstitute parkin-dependent ubiquitination of Mfn2 on the OMM of these isolated mitochondria by adding the E1, E2 and E3 (parkin) components of this pathway, as well as Ub and other factors, as previously described (*Tang et al., 2017*), in a time-, depolarization- and ligase-dependent manner (*Figure 8C*). Using depolarized mitochondria isolated from cells depleted of PINK1 (*Figure 8D*), Mfn2 ubiquitination was almost completely abolished (*Figure 8E*), demonstrating an as-expected requirement for PINK1 in parkin-dependent ubiquitination.

Although we observed robust Mfn2 (and Mfn1) ubiquitination in reactions with depolarized mitochondria and WT parkin, we observed very little to no ubiquitination of other OMM substrates, such as VDAC1, HK1 or TOM20 (*Figure 8F*, compare with *Figure 2A and B*). Based on our data in cells, we reckoned that a dearth of p97 in this *in organello* system may prohibit modification of parkin substrates downstream of Mfn2. We first addressed this by isolating cytosol ('S200k') from mouse brain – which was devoid of mitochondrial, ER and endosomal markers (*Figure 8G*) – to use as a source of cytosolic p97 ATPase (*Otter-Nilsson et al., 1999*). As parkin itself is cytosolic (*Figure 8G*), we initially proceeded to co-incubate *in organello* ubiquitination reactions with cytosol from parkin$^{+/+}$ ('WT cytosol') and parkin$^{-/-}$ ('KO cytosol') mouse brain in the absence of recombinant ligase, and observed that cytosolic, mouse parkin was able to catalyze Mfn2 ubiquitination in a depolarization-dependent manner, albeit not to the extent of 100 nM recombinant GST-parkin (*Figure 8H*; here

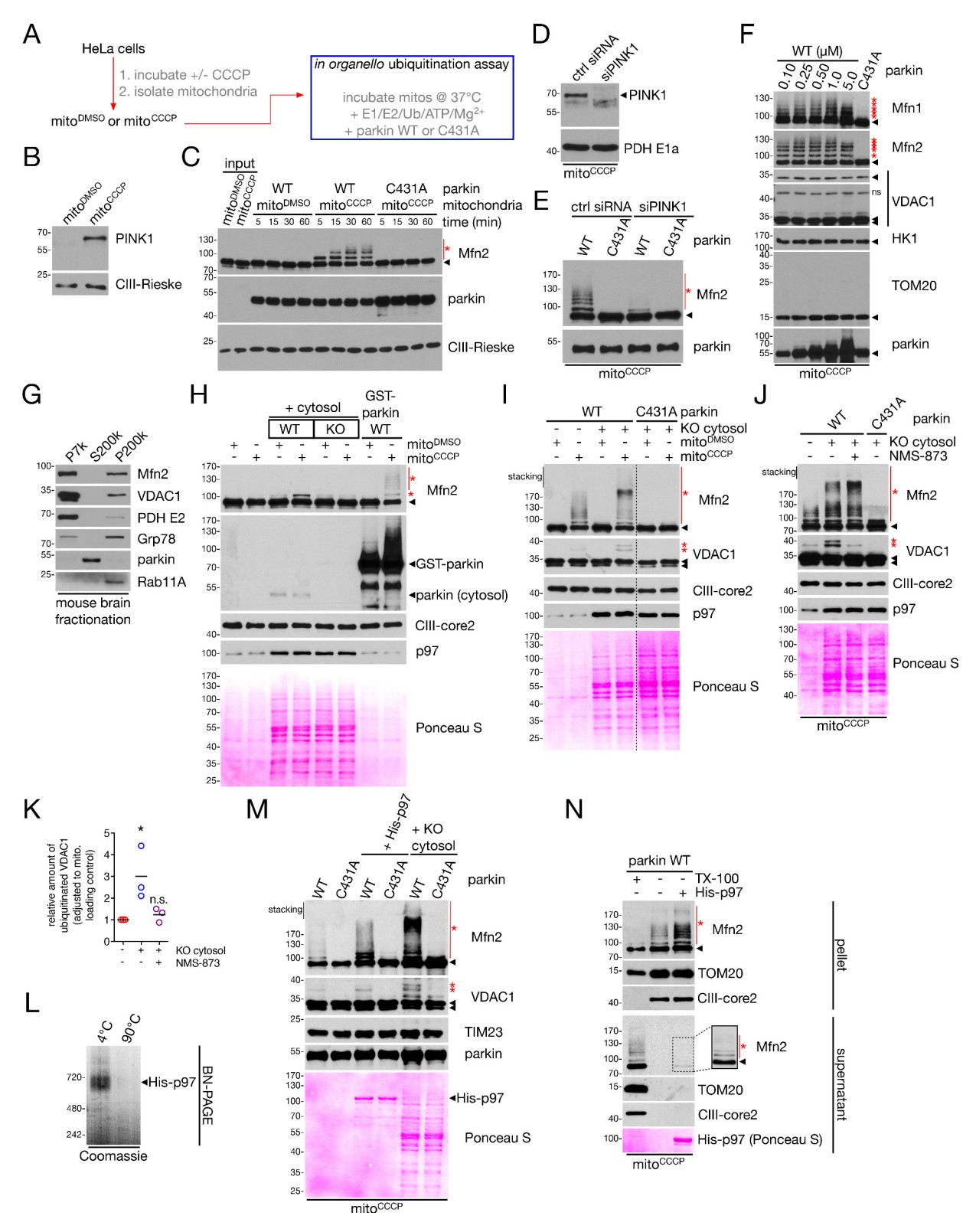

**Figure 8.** In organello ubiquitination of Mfn2 and VDAC1. (**A**) Workflow for the *in organello* ubiquitination assay, where HeLa cells are depolarized with 20 μM CCCP for four hours and mitochondria are isolated ('mito^CCCP', with control 'mito^DMSO'). These are combined with ubiquitination assay components (blue box) and incubated at 37°C (see *Materials and Methods* for full details). (**B**) Immunoblot analysis of PINK1 levels in mitochondria isolated from depolarized ('mito^CCCP') or control ('mito^DMSO') cells. (**C**) *In organello* ubiquitination assays, using depolarized or control mitochondria and

*Figure 8 continued on next page*

*Figure 8 continued*

100 nM of the indicated parkin construct, were incubated at 37°C for the indicated time, and reactions were quenched with SDS-PAGE sample buffer. Mfn2 ubiquitination was analyzed by immunoblot. Ubiquitinated species are indicated by red asterisks, while unmodified bands are denoted by arrowheads. (D) Immunoblot analysis of PINK1 levels in mitochondria isolated from depolarized cells transfected with control siRNA (ctrl siRNA) or siRNA targeting PINK1 (siPINK1). (E) Mitochondria from (D) were used for 30 min *in organello* ubiquitination assays using 100 nM WT or C431A parkin, and Mfn2 ubiquitination was analyzed by immunoblot. Ubiquitinated species are indicated by red asterisks, while unmodified bands are denoted by arrowheads. (F) Depolarized mitochondria were used for 30 min *in organello* ubiquitination assays with the indicated concentration of WT parkin, or 100 nM parkin C431A as a negative control. Ubiquitinated species are indicated by red asterisks, while unmodified bands are denoted by arrowheads. (G) Immunoblot analysis of mouse brain fractionation. Mouse brain homogenate was separated into heavy membrane (P7k), cytosolic (S200k) and light membrane (P200k) fractions. Distribution of mitochondrial (Mfn2, VDAC1, PDH E2), ER (Grp78), soluble (parkin) and endosomal (Rab11A) markers are shown. (H) CCCP-uncoupled ('mito^CCCP') or control ('mito^DMSO') mitochondria were incubated for 60 min with 2 mg/ml cytosol from WT mouse brain ('WT cytosol') or from the brain of parkin^-/- mice ('KO cytosol'). As a positive control, mitochondria were incubated with 100 nM uncleaved GST-parkin^WT (without cytosol). Ubiquitinated species are indicated by red asterisks, while unmodified bands are denoted by arrowheads. (I) CCCP-uncoupled ('mito^CCCP') or control ('mito^DMSO') mitochondria were incubated for 60 min with 100 nM parkin WT or C431A and in the presence or absence of 2 mg/ml cytosol from parkin^-/- mouse brain ('KO cytosol'). Mfn2 and VDAC1 ubiquitination were assayed by immunoblot. Ubiquitinated species are indicated by red asterisks, while unmodified bands are denoted by arrowheads. (J) *In organello* ubiquitination reactions were performed with parkin^-/- mouse brain ('KO cytosol') in the presence of absence of 25 µM NMS-873. Reactions were incubated on ice for 30 min prior to a 60 min 37°C incubation. In the immunoblot analysis, ubiquitinated species are indicated by red asterisks, while unmodified bands are denoted by arrowheads. (K) Quantification of the level of ubiquitinated VDAC1 as compared to control, relative to mitochondrial loading control (TIM23 or CIII-core2). Data points are represented on the graph, n = 3 experiments. *, $p < 0.05$; n.s., not significant. (L) Recombinant, hexameric His-p97 runs as a ~ 700 kDa complex as assayed by BN-PAGE. Prior to separation on the gel, samples were incubated at the indicated temperature for 10 min. (M) Immunoblot analysis of 60 min *in organello* ubiquitination assays using depolarized mitochondria, 100 nM parkin, 200 nM His-p97 hexamer, and 2 mg/ml parkin KO brain cytosol. Ubiquitinated species are indicated by red asterisks, while unmodified bands are denoted by arrowheads. Recombinant His-p97 is additionally indicated on the Ponceau. (N) *In organello* retrotranslocation of Mfn2. *In organello* ubiquitination reactions with or without recombinant p97 were centrifuged at 10,000 g to separate mitochondria (pellet) from soluble factors (supernatant). As a control, reactions were lysed in 1% TX-100 prior to centrifugation. The inset on the supernatant Mfn2 blot shows ubiquitination (red asterisks) of the protein at higher exposure.

DOI: https://doi.org/10.7554/eLife.32866.024

The following source data and figure supplement are available for figure 8:

**Source data 1.** Numerical source data for *Figure 8K*.
DOI: https://doi.org/10.7554/eLife.32866.026

**Figure supplement 1.** Effect of NMS-873 on cytosol-free ubiquitination.
DOI: https://doi.org/10.7554/eLife.32866.025

the GST tag was not cleaved in order to visualize the different forms of parkin by immunoblot). Based on this result, we proceeded to co-incubate isolated mitochondria with KO cytosol and recombinant parkin. Under these conditions, we observed robust ubiquitination of both Mfn2 and VDAC1 compared to reactions lacking KO cytosol (*Figure 8I*). This result indicated a potential role for p97 (which was present in the cytosol, *Figure 8H and I*) in this process and, remarkably, in cytosol-containing reactions, VDAC1 ubiquitination displayed sensitivity to NMS-873 (*Figure 8J and K*). Importantly, NMS-873 had no effect on ubiquitination in the absence of cytosol (*Figure 8—figure supplement 1*), indicating that the small amount of p97 present in reactions lacking cytosol was either negligible or already engaged with other substrates. To ensure that p97 was truly stimulating VDAC1 ubiquitination, we added recombinant p97 hexamer to our reactions (*Figure 8L*). The addition of recombinant hexamer, in a parkin-dependent manner, stimulated both Mfn2 and VDAC1 ubiquitination (*Figure 8M*). This was, however, not to the extent seen with cytosol (which is p97-dependent, *Figure 8J and K*), as other cytosolic factors, notably p97 cofactors and E4 ligases, are also likely involved.

Finally, we tested whether retrotranslocation of Mfn2 by recombinant p97 was occurring in our *in organello* reactions. By fractionating samples post-reaction into mitochondria (pellet) and soluble factors (supernatant), we observed a small amount of Mfn2 appear in the supernatant only when recombinant p97 was added to the ubiquitination reaction (*Figure 8N*; samples were solubilized with TX-100 as a positive control). A longer exposure revealed that retrotranslocated Mfn2 was indeed ubiquitinated (box in *Figure 8N*). Taken together, our *in organello* ubiquitination data show that, in a cell-free assay, we can reconstitute PINK1/parkin-dependent, p97-stimulated Mfn2 and VDAC1 ubiquitination, and Mfn2 retrotranslocation. These results are in line with our experiments in cells which demonstrate that PINK1, parkin and p97 collaborate to uncouple OMM-ER contacts via

Mfn2 ubiquitination and degradation during mitophagy, which in turn allows ubiquitination and degradation of additional parkin substrates such as VDAC1.

## Discussion

Here, we have described a reciprocal relationship between mitochondria-ER tethering and mitophagy. Contacts between both organelles are destroyed during mitophagy, in both heterologous cell cultures and dopaminergic neurons, and we demonstrate a requirement for parkin, PINK1, p97 and proteasomal activity in this process. Complementarily, mitochondria-ER contacts themselves are negative regulators of mitophagy, as their reduction facilitates parkin substrate ubiquitination, its translocation to mitochondria and mitochondrial turnover. We identify the known mitochondria-ER tether Mfn2 as a factor that is rapidly phosphoubiquitinated upon the induction of mitophagy, and show that Mfn2-containing HMW complexes are extracted from the OMM by p97 in a manner requiring parkin-dependent ubiquitination in the Mfn2 HR1 domain. Both reduction of Mfn2 ubiquitination and p97 inhibition repress mitophagy, and we reconstitute the main concepts of this PINK1/parkin/p97 enzymatic system in a cell-free assay. Overall, we identify a regulatory role for Mfn2-mediated mitochondria-ER coupling within the parkin/PINK1 pathway, which is counteracted by the ubiquitination of Mfn2 by parkin and its p97-dependent proteasomal turnover.

We propose a model in which the PINK1/parkin/p97 axis acts rapidly on Mfn2 HMW complexes to separate mitochondria from the ER in order to facilitate mitophagy, potentially by making more substrates available to the parkin/PINK1 system (*Figure 9*). Emerging from this model is the intriguing possibility that mitochondria-ER contacts are initial sites of PINK1/parkin activity and Ub phosphorylation, and would thus be critical loci of mitophagic regulation by deubiquitinating enzymes and as-yet unidentified ubiquitin phosphatases. A recent cryoelectron tomographical study on the ancestral yeast mitofusin Fzo1p demonstrated the existence of a ring-like structure formed by Fzo1p during the docking stage of mitochondrial fusion (*Brandt et al., 2016*). Mfn2 bridges between mitochondria and the ER may therefore form a similar type of ring, potentially restricting the availability of non-mitofusin OMM substrates such as VDAC1 (*Figure 7C–H*, and *Figure 8I–M*) to parkin and/or PINK1. With respect to the latter case, PINK1 has recently been shown to localize to the mitochondria-associated membrane of the ER (MAM) upon depolarization (*Gelmetti et al., 2017*), and a physical interaction between VDACs on the OMM and IP$_3$ receptors on the ER places this parkin substrate at contacts between both organelles (*Szabadkai et al., 2006*). The existence of a ~ 500 kDa Mfn2-containing interorganellar bridge is supported by our BN-PAGE data (*Figure 5A*) demonstrating that Mfn2 uniquely exists in a homotypic dimer of ~250 kDa subunits, as it has been demonstrated that ~500 kDa mitofusin complexes form from subcomplexes on adjacent membranes (*Ishihara et al., 2004*). Our observation of a steady-state ~500 kDa complex containing Mfn2 but not Mfn1 correlates with the reduced activity of the Mfn2 GTPase domain in comparison to Mfn1 (*Ishihara et al., 2004*), supports a distinct role for Mfn2 in OMM-ER tethering (*Figure 3—figure supplement 1* and [*de Brito and Scorrano, 2008*]), and fits a model in which Mfns tether membranes in the GTP-bound state (*Brandt et al., 2016*; *Ishihara et al., 2004*; *Qi et al., 2016*). Here, we show that the stability of these complexes can be negatively regulated by parkin-mediated Mfn2 ubiquitination crucially coupled to p97-dependent retrotranslocation. Intriguingly, we observed both ubiquitinated and unmodified forms of retrotranslocated Mfn2 upon p97 addition (*Figure 8N*). This may hint that, while the hexamer engages directly with Mfn2 at the high concentrations used in our assay, Ub-binding cofactors may localize the hexamer to ubiquitinated Mfn2 at physiological levels of p97. Indeed, in ER-associated degradation, p97 recognizes both Ub-dependent and intrinsic signals (*Ye et al., 2003*). The above findings, taken together with another study demonstrating that MITOL-mediated Mfn2 ubiquitination (on different lysine residues) can positively regulate complex formation and mitochondria-ER tethering (*Sugiura et al., 2013*), emphasize Mfn2 ubiquitination as an important regulator of mitochondria-ER contact.

Robust parkin activation during mitophagy occurs through a feed-forward mechanism (*Ordureau et al., 2014*). PINK1-phosphorylated Ub serves to both activate and anchor parkin to the OMM, where it can ligate more Ub moieties that are subsequently phosphorylated (*Okatsu et al., 2015*; *Ordureau et al., 2014*). Here, our data hint at a hierarchy of parkin substrates. The Mfns undergo a burst of phosphoubiquitination at the onset of mitophagy, driven by localized parkin activation – potentially due to their proximity to PINK1 (*Chen and Dorn, 2013*). Indeed, our GFP-parkin

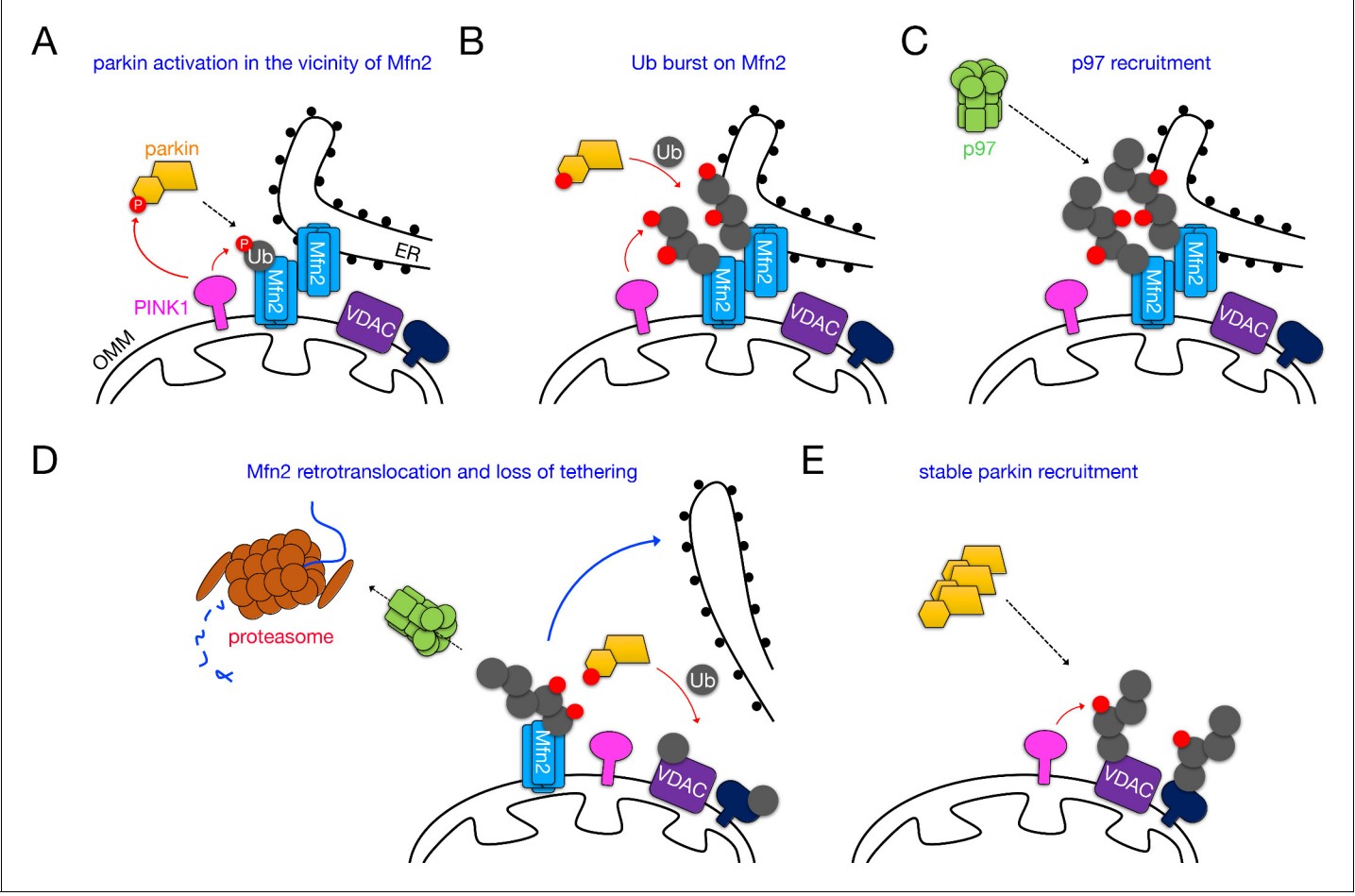

**Figure 9.** Dismantling of Mfn2 interorganellar bridges by PINK1, parkin and p97 during mitophagy. (**A**) PINK1-phosphorylated Ub on Mfn2 initially recruits parkin to Mfn2 complexes, where it is phosphorylated and activated by PINK1. (**B**) Parkin and PINK1 cooperate to catalyze a pUb burst on Mfn2. (**C**) Ubiquitinated Mfn HMW complexes are recognized by p97, which translocates to mitochondria. (**D**) Ubiquitinated Mfn2 is retrotranslocated from the OMM and degraded by the proteasome. (**E**) VDACs and possibly other substrates become available to the parkin/PINK1 system, and their phosphoubiquitination stabilizes parkin on mitochondria to drive mitophagy.

DOI: https://doi.org/10.7554/eLife.32866.027

immunoprecipitation (*Figure 2C*), OMM substrate turnover kinetics (*Figure 2A*), and reconstitution assays (*Figure 8F*) clearly show a preference for the Mfns above other parkin substrates such as HK1 and Miro1. The Mfns are then rapidly extracted from the OMM by p97 (*Figure 5A and E*, *Figure 8N*) in a step that coincides temporally with parkin translocation to mitochondria. It is therefore unlikely that Mfn1 or Mfn2 act as a parkin receptor in this paradigm – as others have suggested (*Chen and Dorn, 2013*) – for this reason, especially when our recruitment data in Mfn2-deficient cells (*Figure 3C–E* and *Figure 3—figure supplement 4*) are taken into account. Indeed, we demonstrate that Mfn2 acts as a stable parkin tether only under conditions where retrotranslocation by p97 is inhibited (*Figure 6J and K*). Our data support a role for the involvement of VDAC1 in a stable complex that tethers parkin to the OMM (*Callegari et al., 2017*); as β-barrel channels fully integrated into the membrane, VDACs may not be amenable to p97-dependent retrotranslocation. Mfn2 may act as a parkin receptor in cardiomyocytes (*Chen and Dorn, 2013*), where parkin-dependent clearance of mitochondria by autophagy plays a role in metabolic development (*Gong et al., 2015*) rather than quality control, and thus may occur by a distinct mechanism; the phosphomutant Mfn2$^{T111A/S442A}$ or Mfn2 deletion blocks parkin-mediated mitophagy in the heart but not in cell lines (*Figure 4J and K*, *Figure 3—figure supplement 4*, and [*Narendra et al., 2008*]). Conceivably, phosphorylation of Mfn2 on T111 and S442 by a cardiac-specific S/T kinase (or cardiac PINK1, as has

been proposed [*Chen and Dorn, 2013*]) may facilitate mitophagy in the heart by uncoupling mitochondria from the sarcoplasmic reticulum.

Our study describes an antagonistic, reciprocal relationship between mitophagy and interorganellar tethering between mitochondria and the ER. This highlights a fundamental difference between mitophagy and the more canonical starvation-induced autophagy pathway, the latter of which requires mitochondria-ER contact sites for autophagosome formation (*Hamasaki et al., 2013*). While mitophagy functions as a quality control mechanism (*Ryan et al., 2015*), starvation-induced autophagy is a metabolic response, and thus its initiation at contact sites between mitochondria and the ER may serve to decode the metabolic needs of the cell. Mechanistically, both mitochondria (*Hailey et al., 2010*) and the ER (*Hayashi-Nishino et al., 2009*) have been reported to function as autophagosomal membrane sources during starvation, and mitochondrial damage may preclude the former from participating in this process during mitophagy. Accordingly, the SNARE Stx17, which governs autophagosome-lysosome fusion during starvation (*Itakura et al., 2012b*), is dispensable for mitophagy (*McLelland et al., 2016*; *Nguyen et al., 2016*). Indeed, Stx17 appears to suppress mitophagy (*Figure 3—figure supplement 3G–I*) through its role in supporting mitochondria-ER contact (*Arasaki et al., 2015*). While mitophagy does indeed share morphological and several mechanistic similarities with canonical macroautophagy – including the recruitment of ULK1 complexes and ATG9A vesicles to depolarized mitochondria (*Itakura et al., 2012a*; *Lazarou et al., 2015*) – molecular dissection of mitophagosome formation and fusion requires further study.

Finally, our data posit the possibility of steady-state regulation of mitochondria-ER contact by PINK1/parkin, separately from mitophagy. In flies, phenotypes of *PINK1* and *PRKN* mutants are duplicated by overexpression of the sole *Drosophila* mitofusin MARF, and are suppressed by p97 overexpression (*Yun et al., 2014*; *Zhang et al., 2017*). Thus, PINK1/parkin/p97 counteract MARF *in vivo* through its ubiquitination and turnover (*Wang et al., 2016*; *Zhang et al., 2017*; *Ziviani et al., 2010*). Indeed, a proposed mechanism of cell death due to deletion of *PINK1* is the sensitization of mitochondria to $Ca^{2+}$ overload (*Akundi et al., 2011*; *Gandhi et al., 2009*; *Kostic et al., 2015*), the root cause of which may be dysregulation of mitochondria-ER contact. Accordingly, deletion of the mitochondrial $Ca^{2+}$ uniporter protects dopaminergic neurons from cell death in *PINK1*-deficient zebrafish (*Soman et al., 2017*). While we did not observe any steady-state differences in the extent of mitochondria-ER coupling in either parkin overexpression (*Figure 1A–E*) or loss-of-function (*Figure 1N and O*) systems, others have observed an increased degree of contact and metabolite transfer in both fibroblasts from *PRKN* and *PINK1* patients, as well as brains from *PINK1* and *PRKN* mutant flies (*Celardo et al., 2016*; *Gautier et al., 2016*). Conversely, we (*Figure 1H–J*) and others (*Gelmetti et al., 2017*) measured a destabilization of mitochondria-ER tethering when PINK1 was transiently depleted. While differences between studies can be attributed to cell type and culture conditions, how mitochondria-ER contact is quantified is certainly a determinant; whereas we quantified ER tubules within 25 to 100 nm of the OMM (*Figure 1* and *Figure 1—figure supplement 1*), Gautier et al. extended this distance to 500 nm, and this may effectively account for observed differences. For this study, our < 100 nm criterion was sufficient to capture ER tubules directly opposite the OMM (see OMM extension outlines in *Figure 1O* and the comparison of ER-OMM distances in *Figure 1—figure supplement 1*). Future work will aim to (a) address when and where PINK1/parkin act to regulate the OMM-ER interface via Mfn2, (b) solve precisely how Mfn2 is recognized and retrotranslocated by p97, and (c) understand how dysregulation of mitochondria-ER contact during mitophagy and in other PINK1/parkin-related paradigms may contribute to disease pathology. The work described here lays the foundation for these future studies, identifying a molecular mechanism for contact site destabilization through the ubiquitination of Mfn2 tethering complexes by the PINK1/parkin system and their extraction and destruction via p97 and the proteasome.

## Materials and methods

**Key resources table**

| Reagent type (species) or resource | Designation | Source or reference | Identifiers | Additional information |
| --- | --- | --- | --- | --- |

*Continued on next page*

*Continued*

| Reagent type (species) or resource | Designation | Source or reference | Identifiers | Additional information |
|---|---|---|---|---|
| cell line (*Homo sapiens*) | U2OS | PMID 24446486 | | |
| cell line (*Hs*) | U2OS:GFP | PMID 24446486 | | |
| cell line (*Hs*) | U2OS:GFP-parkin WT | PMID 24446486 | | |
| cell line (*Hs*) | U2OS:GFP-parkin A320R | PMID 28276439 | | |
| cell line (*Hs*) | Mfn2 KO | this paper | | see Plasmids and transfection |
| cell line (*Hs*) | Mfn2 KO:YFP-parkin WT | this paper | | see Plasmids and transfection |
| cell line (*Hs*) | Mfn2 KO:YFP-parkin C431S | this paper | | see Plasmids and transfection |
| cell line (*Hs*) | HeLa | PMID 24446486 | | |
| cell line (*Hs*) | control-1 | NIH | NCRM-1 | |
| cell line (*Hs*) | control-2 | PMID 27641647 | | |
| cell line (*Hs*) | *PRKN*(del) | PMID 20885945 | | |
| transfected construct (*Hs*) | HA-Ub | PMID 25216678 | | |
| transfected construct (*Hs*) | DsRed2-LC3 | PMID 18596167 | | |
| transfected construct (*Hs*) | Mfn1-HA | PMID 15878861 | | |
| transfected construct (*Hs*) | Mfn2 WT | PMID 15878861 | | |
| transfected construct (*Hs*) | Mfn2 K406R | this paper | | see Plasmids and transfection |
| transfected construct (*Hs*) | Mfn2 K416R | this paper | | see Plasmids and transfection |
| transfected construct (*Hs*) | Mfn2 K420R | this paper | | see Plasmids and transfection |
| transfected construct (*Hs*) | Mfn2 HR1 | this paper | | see Plasmids and transfection |
| transfected construct (*Hs*) | Mfn2 HR2 | this paper | | see Plasmids and transfection |
| transfected construct (*Hs*) | Mfn2 TS/AA | this paper | | see Plasmids and transfection |
| transfected construct (*Hs*) | GFP-parkin WT | PMID 24446486 | | |
| biological sample (*Mus musculus*) | parkin WT brain cytosol | this paper | | see In organello ubiquitination assays |
| biological sample (*Mm*) | parkin KO brain cytosol | this paper | | see In organello ubiquitination assays |
| antibody | anti-actin | Millipore | MAB1501 | |
| antibody | anti-B-III-tubulin | Sigma | T8660 | |
| antibody | anti-MAVS | Enzo | ALX-210–929 C100 | |
| antibody | anti-cytochrome c | BD | 556432 | |
| antibody | anti-GFP | Abcam | ab6673 | IP |
| antibody | anti-GFP | Invitrogen | A6455 | WB |
| antibody | anti-Grp78 | Santa Cruz | sc-376768 | |
| antibody | anti-HA | Abcam | ab9134 | |

*Continued on next page*

*Continued*

| Reagent type (species) or resource | Designation | Source or reference | Identifiers | Additional information |
|---|---|---|---|---|
| antibody | anti-HK1 | Cell Signaling | 2024S | |
| antibody | anti-Mfn1 | Santa Cruz | sc-50330 | |
| antibody | anti-Mfn2 | Sigma | M6319 | WB in *Figure 3—figure supplement 2D* |
| antibody | anti-Mfn2 | Cell Signaling | 9482 | all other assays (IF, WB, IP) |
| antibody | anti-CIV-COXI | Abcam | ab14705 | |
| antibody | anti-p62 | Progen | GP62-C | |
| antibody | anti-PDH E1a | Abcam | ab110330 | |
| antibody | anti-PDH E2/E3bp | Abcam | ab110333 | |
| antibody | anti-PDI | Abcam | ab2792 | |
| antibody | anti-PINK1 | Cell Signaling | 6946 | |
| antibody | anti-pS65 Ub | Millipore | ABS1513-I | |
| antibody | anti-Rab11A | Cell Signaling | 2413 | |
| antibody | anti-Miro1 | Sigma | HPA010687 | |
| antibody | anti-CII-SDHA | Abcam | ab14715 | |
| antibody | anti-Stx17 | ProteinTech | 17815–1-AP | |
| antibody | anti-TH | Pel-Freez | P40101-150 | |
| antibody | anti-TIM23 | BD | 611222 | |
| antibody | anti-TOM20 | Santa Cruz | sc-11414 | |
| antibody | anti-TOM70 | Santa Cruz | sc-390545 | |
| antibody | anti-Ub [FK2] | Enzo | BML-PW8810 | IF |
| antibody | anti-Ub [P4D1] | Santa Cruz | sc-8017 | WB |
| antibody | anti-CIII-core2 | Abcam | ab14745 | |
| antibody | anti-CIII-Rieske | Abcam | ab14746 | |
| antibody | anti-p97 | Abcam | ab11433 | |
| antibody | anti-VDAC1 | Abcam | ab14734 | |
| recombinant protein (*Rattus norvegicus*) | GST-R0RBR WT | PMID 23661642 | | |
| recombinant protein (*Rn*) | GST-R0RBR A320R | this paper | | see Plasmids and transfection |
| recombinant protein (*Rn*) | GST-parkin WT | PMID 28276439 | | |
| recombinant protein (*Rn*) | GST-parkin C431A | PMID 28276439 | | |
| recombinant protein (*Hs*) | UbcH7 | PMID 28276439 | | |
| recombinant protein (*Hs*) | UBE1 | BostonBiochem | E-305 | |
| recombinant protein (*Hs*) | Ubiquitin | BostonBiochem | U-100H | |
| recombinant protein (*Hs*) | Usp2 catalytic domain | BostonBiochem | E-504 | |
| recombinant protein (*Tribolium castaneum*) | *Tc*PINK1 | PMID 24784582 | | |
| recombinant protein (*Hs*) | GST-4xUb G76V | PMID 23670163 | | |
| recombinant protein (*Mm*) | His-p97 | PMID 19506019 | | |

*Continued on next page*

Continued

| Reagent type (species) or resource | Designation | Source or reference | Identifiers | Additional information |
|---|---|---|---|---|
| commercial assay or kit | QuikChange II site-directed mutagenesis kit | Agilent | 200523 | |
| commercial assay or kit | BCA protein assay | ThermoFisher | 23227 | |
| chemical compound, drug | CCCP | Sigma | C2759 | |
| chemical compound, drug | MG132 | Sigma | M8699 | |
| chemical compound, drug | Hoechst 33342 | ThermoFisher | H3570 | |
| chemical compound, drug | NMS-873 | ApexBio | B2168 | |
| software, algorithm | BioTools | Bruker | | |
| software, algorithm | MASCOT | Matrix Science | | |
| software, algorithm | Data Analysis | Bruker | | |
| software, algorithm | ImagJ | NIH | | |
| software, algorithm | PyMOL | Schrodinger | | |
| software, algorithm | Excel | Microsoft | | |
| software, algorithm | Prism | GraphPad | | |

## Antibodies and other reagents

Antibodies used in this study include anti-actin (Millipore, MAB1501), anti-β-III tubulin (Sigma-Aldrich, T8660), anti-Cardif (referred to herein as MAVS, Enzo Life Sciences, ALX-210–929 C100), anti-cytochrome c (BD Biosciences, 556432), anti-GFP (ab6673, Abcam), anti-GFP (A6455, Invitrogen), anti-Grp78 (Santa Cruz, sc-376768), anti-HA (Abcam, ab9134), anti-HK1 (Cell Signaling Technology, 2024S), anti-Mfn1 (Santa Cruz, sc-50330), anti-Mfn2 (Sigma-Aldrich, M6319), anti-Mfn2 (Cell Signaling, 9482), anti-MTCO1 (herein referred to as CIV-COXI, ab14705), anti-p62 (Progen, GP62-C), anti-PDH E1a (Abcam, ab110330), anti-PDH E2/E3bp (Abcam, ab110333), anti-PDI (Abcam, ab2792), anti-PINK1 (Cell Signaling, 6946), anti-pS65 ubiquitin (Millipore, ABS1513-I), anti-Rab11A (Cell Signaling, 2413), anti-Rhot1 (referred to herein as Miro1, Sigma-Aldrich, HPA010687), anti-SDHA (referred to herein as CII-SDHA, Abcam, ab14715), anti-Stx17 (ProteinTech, 17815–1-AP), anti-TH (Pel-Freez, P40101-150), anti-TIM23 (BD, 611222), anti-TOM20 (Santa Cruz, sc-11414), anti-TOM70 (Santa Cruz, sc-390545), anti-ubiquitin [FK2] (Enzo Life Sciences, BML-PW8810), anti-ubiquitin [P4DI] (Santa Cruz, sc-8017), anti-UQCRC2 (referred to herein as CIII-core2, Abcam, ab14745), anti-UQCRFS1 (referred to herein as CIII-Rieske, Abcam, ab14746), anti-VCP (referred to herein as p97, Abcam, ab11433) and anti-VDAC1 (Abcam, ab14734). Halt phosphatase inhibitor cocktail was purchased from Thermo Fisher Scientific, and NMS-873 was purchased from ApexBio. Unless otherwise specified, all other reagents were purchased from Sigma-Aldrich.

## Cell culture and iPS cell differentiation

U2OS and HeLa cells were purchased from ATCC, tested negative during routine tests for mycoplasma contamination, and were maintained in DMEM supplemented with L-glutamine, penicillin/streptomycin, and 10% FBS in the presence of either 25 mM glucose or 10 mM galactose (Wisent, Saint-Bruno, QC). Glucose-maintained cells were shifted to galactose-containing medium for at least seven days before use in experiments. The parkin mutant iPSC line (*PRKN*del) was initially isolated from a patient carrying compound heterozygous deletions (delEx7/c.1072delT) in the *PRKN* gene (*Grünewald et al., 2010*). Control lines used in this study were NCRM1 (NIH, Bethesda, MD) and L2131 (*Chung et al., 2016*). Differentiation of iPSCs into dopaminergic neurons was based on a protocol by Xi and colleagues (*Xi et al., 2012*). iPSCs were initially grown in non-coated flasks for one week in DMEM/F12 supplemented with N2 and B27, in the presence of 10 μM SB431542, 200 ng/ml

noggin, 1 µM CHIR99021, 200 ng/ml Shh and 100 ng/ml FGF-8. Embryoid bodies were transferred to polyornithine- and laminin-coated flasks to form rosettes, grown in the presence and then absence of the above-indicated differentiation factors for one week each. Neural progenitors were then cultured in 50% DMEM/F12 and 50% Neurobasal medium, supplemented with N2 and B27, in the presence of 1 µg/ml laminin, 500 µM db-cAMP, 20 ng/ml BDNF, 20 ng/ml GDNF, 200 µM ascorbic acid, 50 µM valproic, 100 nM Compound A and 1 ng/ml TGF-β. Progenitors were then grown in 25% DMEM/F12 and 75% Neurobasal medium, supplemented as above, for three days, and final differentiation into dopaminergic neurons occurred over four weeks in Neurobasal medium (supplemented as above).

## Plasmids and transfection

Cells were transfected with siRNA or DNA using jetPRIME transfection reagent (Polyplus Sciences) according to the manufacturer's instructions. Cells were typically analyzed three or one day(s) after siRNA or DNA transfection, respectively. The codon-optimized GST-R0RBR (*Trempe et al., 2013*), DsRed-LC3 (*Boland et al., 2008*), HA-Ub (*Durcan et al., 2014*), His-p97 (*Halawani et al., 2009*) and Mfn2 (*Neuspiel et al., 2005*) plasmids have been described previously. Mfn mutants were generated using the QuikChange II site-directed mutagenesis kit (Agilent Technologies) according to the manufacturer's instructions and confirmed by sequencing. While duplexed oligonucleotides were used in the mutagenesis reactions, only forward primers are listed below. Mfn2$^{HR1}$ was created by sequential reactions with 5'-CTGAAATTTATTGACAGACAGCTGGAGCTCTTG-3' and 5'-CTTGGCTCAAGACTA TAGGCTGCGAATTAAGCAG-3' to create Mfn2$^{K406R/K416R}$, then with 5'-CTATAGGCTGCGAA TTAGGCAGATTACGGAGGAAG-3' to make Mfn2$^{HR1}$, as this last primer contains the K416R substitution already present. Likewise, Mfn2$^{HR2}$ was created by sequential reactions with 5'-CCGCCA TGAACAAGAGAATTGAGGTTCTTG-3', 5'-CTCACTTCAGAGCAGAGCAAAGCTGCTC-3' and 5'-C TGCTCAGGAATAGAGCCGGTTGGTTG-3' to make Mfn2$^{K720R/K730R/K737R}$, and then with 5'-GCCGCCATGAACAGGAGAATTGAGGTTC-3' to make the final K719R mutation. Mfn2$^{T111A/S442A}$ was created using 5'-CAATGGGAAGAGCGCCGTGATCAATGC-3' and 5'-GAGGAGATCAGGCGCC TCGCAGTACTGGTGGACGATTAC-3'. GST-R0RBR$^{A320R}$ was created using 5'-ACCAGCAGTACGG TCGTGAAGAATGCGTTCTG-3'. U2OS:GFP, U2OS:GFP-parkin$^{WT}$, U2OS:GFP-parkin$^{C431S}$ and U2OS: mtKeima stable cell lines have been described previously (*Tang et al., 2017*), and the Mfn2 KO:YFP-parkin$^{WT}$ and Mfn2 KO:YFP-parkin$^{C431S}$ lines were created in the same manner using YFP-parkin constructs generated in that study. To create the initial Mfn2 KO U2OS cell lines, the human *MFN2* gene was disrupted in exon three using the following guide RNA: 5'-CACUUAAGCACUUUGUCAC U-3'. To create the GST-4xUb$^{G76V}$ construct, the 4xUb fragments from pCMV-TOM70-2xFLAG-4xUb (*Zheng and Hunter, 2013*) were subcloned by digestion with BamHI and XhoI and ligation into pGEX6P1. This Ub chain is composed of four tandem copies of ubiquitin G76V, which mimic a linear Ub chain but cannot be cleaved in the cell by the Ub processing machinery. siRNA targeting p97, PINK1 and Stx17 have been previously described (*McLelland et al., 2016*; *McLelland et al., 2014*). Non-targeting siRNA oligonucleiotides, as well as siRNA targeting Mfn1 (5'-GAUACUAGCUACUG UGAAAdTdT-3') (*Zhao et al., 2013*), Mfn2 (5'-GGAAGAGCACCGUGAUCAAdTdT-3') (*Zhao et al., 2013*) and PACS2 (5'-AACACGCCCGUGCCCAUGAACdTdT-3') (*Simmen et al., 2005*) were purchased from Thermo Fisher Scientific.

## Cell lysis and immunoblotting

Cells were lysed in lysis buffer (20 mM Tris pH 7.5, 150 mM NaCl, 1 mM EDTA, 1 mM EGTA, 1% NP-40 substitute, 1% sodium deoxycholate, protease inhibitor cocktail [aprotinin, leupeptin and benzamidine], and phosphatase inhibitor cocktail) on ice. Lysates were cleared by centrifugation, protein was quantified by BCA assay (Pierce/Thermo Scientific), separated by SDS-PAGE over Tris-glycine gels and transferred to nitrocellulose membrane. Primary antibodies were diluted in 3% BSA in PBS-Tween and incubations performed overnight at 4°C. The following day, membranes were washed and incubated in HRP-conjugated secondary antibodies (Jackson ImmunoResearch Laboratories), diluted in 5% milk in PBS-Tween, at room temperature for one hour. Protein bands were detected using Western Lightning ECL and Plus-ECL kits (PerkinElmer), according to the manufacturer's instructions.

## Immunoprecipitation

Cells were lysed in HEPES-IP buffer (20 mM HEPES pH 7.2, 150 mM NaCl, 1% NP-40 substitute, 0.1% sodium deoxycholate, and protease/phosphatase inhibitor cocktails) and protein content was quantified by BCA assay after clearing by centrifugation. For immunoprecipitation under denaturing conditions, cells were alternatively lysed in 10 mM Tris pH 7.4, 1% SDS, 5 mM EDTA, 10 mM DTT and protease/phosphatase inhibitor cocktails and incubated for 10 min at 90°C. Post-lysis, nine volumes of 10 mM Tris pH 7.4, 150 mM NaCl, 1% Triton X-100, 1 mM EDTA, 1 mM EGTA and protease/phosphatase inhibitor cocktails were added to the sample, and then protein was quantified. Lysates were equilibrated to 1 to 2 mg/ml protein and immunocapture was performed with the indicated antibody overnight at 4°C at a 1:10 to 1:100 dilution. The following day, immunoprecipitation was performed with protein A- or protein G-sepharose (GE Healthcare) for four hours at 4°C. Immunoprecipitates were washed five times in buffer and eluted by incubating in SDS-PAGE sample buffer at 90°C.

## Mitochondrial isolation and BN-PAGE

After treatment, U2OS cells were collected from $2 \times 15$ cm plates per condition in isolation buffer (20 mM Hepes pH 7.4, 220 mM mannitol, 68 mM sucrose, 76 mM KCl, 4 mM KOAc, and 2 mM MgCl$_2$, supplemented with protease inhibitors benzamidine, PMSF, aprotinin, and leupeptin) and passed through a 27.5-gauge syringe twenty times. Cell lysates were centrifuged at 600 g for 10 min at 4°C. Supernatants were then centrifuged at 10,000 g for 10 min at 4°C. The mitochondrial pellet was resuspended in isolation buffer and centrifuged again at 12,000 g for 10 min at 4°C. Protein content of mitochondria was determined by BCA assay, and equilibrated to 1 mg/ml prior to lysis with 1% NP-40 substitute at 4°C for 30 min. Mitochondrial lysates were clarified by centrifugation and added to sample buffer and Coomassie Blue G-250. Solubilized complexes were separated over 4–16 and 3–12% Bis-Tris gels and transferred to PVDF membrane using the NativePAGE Novex Bis-Tris gel system (Life Technologies) according to the manufacturer's instruction prior to immunoblotting. In addition, certain samples were incubated with 1 µM Usp2 (Boston Biochem) for 30 min at 37°C following NP-40 lysis, then separated by SDS-PAGE as above.

## In organello ubiquitination assays

*In organello* ubiquitination was performed as previously described (*Tang et al., 2017*). HeLa cells were depolarized with 20 µM CCCP (or DMSO control) for 4 hr, and then mitochondria were isolated in isolation buffer as described in the previous section. Isolated mitochondria were incubated (at a final concentration of 0.5 to 1.0 mg/ml) with 20 nM E1 Ub activating enzyme, 100 nM UbcH7, 5 µM Ub, 4 mM ATP, 5 mM MgCl$_2$, 50 µM TCEP and (unless otherwise indicated) 100 nM parkin at 37°C for the indicated time (typically 30–60 min, vortexing at 15 min intervals), then quenched in SDS-PAGE sample buffer. E1 enzyme and Ub were purchased from Boston Biochem (Cambridge, MA). In certain cases, reactions were co-incubated with 200 nM His-p97 hexamer or 2 mg/ml mouse brain cytosol. Purification of murine His-p97 has been described previously (*Halawani et al., 2009*). Additionally, isolation of cytosol from mouse tissue (200,000 g supernatant) was performed as previously described (*McLelland et al., 2016*). SDS-PAGE sample buffer was then added to pellets and supernatants prior to SDS-PAGE and immunoblot analysis.

## LC/MS on immunoprecipitated Mfn2

Mfn2 immunoprecipitates were washed twice in PBS, then twice more in 50 mM ammonium acetate pH 7.0, and eluted twice in 50% acetic acid on ice for 10 min. Eluates were pooled, cleared by centrifugation and dried by speedvac. Pellets were resuspended in 8 µl 6 M urea, 50 mM TEAB pH 8.5, and diluted with 40 µl 50 mM TEAB pH 8.5. The sample was reduced in 2 mM TCEP at 37°C for ten minutes, and then alkylated in 20 mM iodoacetamide for 30 min at room temperature in the dark. The 50 µl sample was then digested with 0.2 µg of trypsin for two hours at 37°C, and then quenched in 0.5% trifluoroacetic acid (TFA) and 5% acetonitrile. Digests were C18-purified using ZipTips (Millipore), eluted in 0.1% TFA/80% acetonitrile, evaporated and resuspended in 0.1% TFA/4% acetonitrile. Peptides were diluted in 0.1% TFA/4% acetonitrile, and eluted from an Acclaim PepMap100 C18 column (75 µm × 25 cm) with a 1 hr 5–40% gradient of acetonitrile in 0.1% formic acid at 300 nL/min. The eluted peptides were analyzed with an Impact II Q-TOF spectrometer equipped with a

Captive Spray nano electrospray source (Bruker). Data were acquired using data-dependent auto-MS/MS with a range 150–2200 m/z range, a fixed cycle time of 3 s, a dynamic exclusion of 1 min, m/z-dependent isolation window (1.5–5 Th) and collision energy 25–75 eV (*Beck et al., 2015*). MS/MS data were analyzed using MASCOT using a search procedure against the SwissProt proteome database (taxonomy: mammalia). The search parameters included a peptide tolerance of 15.0 ppm, an MS/MS tolerance of 0.05 Da, up to two $^{13}$C atoms per peptide, up to two missed trypsin cleavage sites, fixed carbamidomethyl and variable methionine oxidation and Ser/Thr phosphorylation modifications. The significance threshold was set to p<0.05. The MASCOT automatic peptide decoy search was performed, and false discovery rates of 1.04 and 1.88% for the DMSO and CCCP-treated samples were obtained, respectively. Only peptides with scores above 18.0 were accepted. For ubiquitin analysis, only one peptide was identified with MASCOT; other Ub peptides were identified by generating tryptic peptides with up to one missed cleavage, and then matched to the LC-MS data with BioTools (Bruker). Peptides with BioTools scores above 10 (Δm/z < 10 ppm for the parent ion and at least 7 MS/MS fragments within less than 0.2 Da) were included in the analysis. Extracted ion chromatograms were integrated using the Data Analysis software (Bruker).

## GST protein purification, in vitro phosphorylation and pulldown

*Tribolium castaneum* PINK1 (*Tc*PINK1, amino acids 128–570), GST-4xUb$^{G76V}$ and GST-R0RBR were expressed as GST fusion proteins in BL21 cells from pGEX6P1 vectors as described (*Koyano et al., 2014*; *Trempe et al., 2013*). Protein expression was induced at 16°C for 16 hr with 100 μM IPTG. After harvesting, the *E. coli* pellet was lysed by sonication in lysis buffer (TBS: 50 mM Tris-HCl pH 8.0, 300 mM NaCl, 1 mM DTT, 0.5% Tween-20 and 5 mM MgSO$_4$), in the presence of lysozyme, DNase I and EDTA-free protease inhibitors. The suspension was centrifuged and the supernatant was applied to Glutathione Sepharose 4B beads (GE Healthcare). After 1 hr of agitation at 4°C the beads were washed with TBS and eluted with TBS containing 20 mM glutathione and 1% CHAPS. GST-*Tc*PINK1 was further cleaved in solution by incubation with GST-3C protease for 12 hr at 4°C before further purification by gel filtration (Superdex 75, GE Life Sciences) in low salt buffer as a final step. GST-4xUb was purified via buffer exchange to remove extra glutathione. GST-4xUb phosphorylation was performed in phosphorylation buffer (50 mM Tris-HCl pH 7.5, 100 m M NaCl, 1 mM ATP, and 5 mM MgSO$_4$) at 30°C for 90 min, at final concentrations of 0.1 mg/ml *Tc*PINK1 and 0.25 mg/ml GST-4xUb. After phosphorylation, the entire reaction was purified with glutathione Sepharose 4B and washed with TBS briefly to remove extra *Tc*PINK1 and ATP. A portion of the beads was boiled in SDS-PAGE sample buffer and separated by Phos-tag gel to monitor the efficiency of phosphorylation. Whole mouse brain was homogenized in 10 mM HEPES pH 7.4, 0.32 M sucrose supplemented with protease inhibitors and phosphatase inhibitors (Roche). The homogenate was centrifuged for 10 min at 1,000 g, and the supernatant was collected. After adding Triton X-100 to a final concentration of 1%, the lysate was rocked at 4°C for 30 min, then centrifuged at 16,200 g for 30 min. The resulting supernatant was used as whole brain lysate for pull-downs. Pull-downs were performed with 50 μg GST-4x(p)Ub bait and 4 mg lysate at 4°C, incubated overnight in the presence of phosphatase inhibitors. After washing with TBS for five times, reactions were resuspended in 50 μl of TBS with 2 μg GST-3C protease and incubated at RT for 4 hr. After separating the beads by centrifugation, the supernatant was prepared for MS sample preparation. The sample was evaporated and then resuspended with 100 mM NH$_4$HCO$_3$ and 5 mM TCEP, then vortexed at 37°C for 30 min. 110 mM chloroacetamide in 100 mM NH$_4$HCO$_3$ was added to a final concentration 55 mM chloroacetamide. Samples were further vortexed for 30 min at 37°C. Trypsin was added to reach an enzyme:protein ratio of 1:50, and digestion was performed overnight. Samples were then dried in a speedvac and reconstituted in 40 μl 0.2% formic acid. Tryptic peptides were loaded on a C18 stem trap from New Objective and separated on a home-made C18 column (15 cm × 150 μm id) at a flow rate of 600 nl/min with a gradient of 5–30% B (A: 0.2% formic acid in water, B: 0.2% formic acid in acetonitrile). The analytical column was coupled to a Q-Exactive Plus (Thermo Fisher Scientific). Resolution was set at 70000 for the survey scan and 17500 for the tandem MS acquisition. A maximum of 12 precursors were sequenced for each duty cycle. AGC target values for MS and MS/MS scans were set to 3e6 (max fill time 50 ms) and 2e4 (max fill time 150 ms), respectively. The precursor isolation window was set to m/z 1.6 with a high energy dissociation normalized collision energy of 25. The dynamic exclusion window was set to 30 s. Tandem mass spectra were searched against the Uniprot human database with carbamidomethylation (C) as fixed modifications, deamidation (NQ) and

oxidation (M) as variable modifications. Tolerance was set at 10 ppm on precursor mass and 0.01 Da on the fragments. The raw data were searched against the Universal Protein Resource (UniProt) (http://www.pir.uniprot.org/) database by using Mascot (http://www.matrixscience.com). Scaffold was used to validate MS/MS based peptide and protein identifications. Peptide identifications were accepted if they could be established at greater than 95.0% probability as specified by the Peptide Prophet algorithm. Protein identifications were accepted if they could be established at greater than 99.0% probability and contained at least two identified peptides. For each group – GST, GST-4xUb$^{G76V}$ and GST-4xpUb$^{G76V}$ – we performed three biological repeats. P-values against the spectrum counts in GST-4xUb pulled-down samples and GST-4xpUb pulled-down samples for each protein were obtained via Fisher's Exact Test built in the Scaffold software. For final analysis of quantified proteins, values were transferred and analysed in Microsoft Excel. The following cut-offs were applied: minimum number of two total spectrum counts; ratio of spectrum counts in the GST pulled down control divided by the total spectrum counts in all samples is lower than 30%. For the GST-R0RBR pulldown, U2OS:GFP-parkin$^{WT}$ cells were treated with 20 µM CCCP for one hour (or left untreated) prior to lysis in 20 mM HEPES pH 7.2, 150 mM NaCl, 1% NP-40 substitute, 0.2% sodium deoxycholate, and protease/phosphatase inhibitor cocktails and then incubated overnight with 10 µg/ml GST-R0RBR WT or A320R on beads (or unconjugated beads as an additional control). The following day, pulldowns were washed five times in lysis buffer and eluted from beads by incubating in SDS-PAGE sample buffer at 90°C.

## Transmission electron microscopy

After treatment, cells grown in chamber slides were fixed in 2.5% glutaraldehyde in PBS for one hour at room temperature, then stored at 4°C overnight before processing. Thin sections on grids were observed in a Tecnai 12 BioTwin transmission electron microscope (FEI) at 120 keV. Images were acquired with a charge coupled device camera (AMT).

## Immunofluorescence and fluorescence microscopy

Cells were grown on glass coverslips, treated then fixed in 6% formaldehyde in PBS for 15 min 37°C. Fixed cells were permeabilized in 0.25% Triton X-100 in PBS for 10 min, and blocked in 10% FBS in PBS. Primary antibodies were diluted in 5% FBS in PBS, and incubations were performed for one hour at room temperature. Incubations with Alexa Fluor-conjugated secondary antibodies (Thermo Fisher Scientific) were performed in the same manner. Cells were counterstained with Hoechst 33342 (Invitrogen) and mounted on glass slides using Aqua Poly/Mount (Polysciences Inc.). Confocal slices (<1 micron-thick) were acquired via a spinning disc confocal microscope (with Andor Yoko-gawa system IX81, Olympus) through a 100X, 1.4 NA or 60X, 1.4 NA objective lens. Widefield microscopy was performed using a Zeiss AxioObserver Z1 microscope through a 63X, 1.4 NA objective lens.

## Fluorescence-activated cell sorting and mtKeima measurements

Quantitative analysis of mitophagy was performed as described previously (*Tang et al., 2017*). U2OS:mtKeima cells were first transfected with siRNA targeting Mfn1 or Mfn2. Two days later, mtKeima was induced with 10 µM ponasterone A, and cells were transfected with GFP-parkin WT or C431S for 12 to 18 hr. The next day (3 days post-siRNA transfection), cells were treated with 20 µM CCCP (or DMSO) for four to five hours, trypsinized and collected in PBS. Cell fluorescence was analyzed by an LSR Fortessa (BD Bioscience) fluorescence-activated cell sorter, using excitation wavelengths of 405 and 561 nm to detect Keima at pH 7.0 and 4.0, respectively, and 488 nm to detect GFP-parkin. Cell fluorescence data were analyzed using FlowJo (Tree Star). For each condition, $10^5$ cells, gated for GFP-parkin expression, were used for the analysis.

## Image and statistical analyses

The numbers of cells quantified per experiment are explicitly indicated in the figure legends. No statistical method was used to predetermine the experimental sample size. Statistical tests and representations of the data were generated using Prism (GraphPad Software, La Jolla, CA). Data are displayed as the mean ± standard error of the mean (SEM). Statistical significance was determined by one- (*Figure 3—figure supplements 1B-D*, *Figure 4F, G, K*, *Figure 8K*) and two-way

(*Figure 1B—E, I, J, N*, *Figure 3E, G*, *Figure 3—figure supplements 2B*, *3D, I, 4F, H, 5C, D, G*, *Figures 4C, 5G, H, 6I, 7B*) ANOVAs followed by Bonferroni post-hoc tests, or one-tailed t-test (*Figure 1—figure supplement 1B*, *Figures 6K* and *7E*). Differences were considered significant if p<0.05. The diagram of the crystal structure of the pUb-parkin complex was created with PyMOL. Images were analyzed using ImageJ (NIH), and analyses were performed blindly.

## Acknowledgements

This work was supported by an operating grant from the Canadian Institutes for Health Research (CIHR) to EAF. GLM was supported by a Canada Graduate Scholarship from the CIHR, and an award from the Montreal Neurological Institute and Desjardins Foundation. We would like to thank Dr Heidi McBride (McGill), Dr Julien Prudent (MRC, Cambridge, UK) and Dr Atsushi Tanaka (Yamagata University, Yamagata, Japan) for stimulating discussions concerning the data presented herein. We are grateful to Dr Matthew Tang for technical advice regarding mtKeima acquisition experiments, stable cell line creation and cell sorting, Dr Marta Vranas for technical help and advice concerning the *in organello* ubiquitination assay, and Dr Adele Tufford for discussions regarding statistical analyses. We also thank Jeannie Mui and the Facility for Electron Microscopy Research (McGill) for TEM sample processing and technical advice. Proteomic analyses for the GST-Ub pulldown were performed by the Centre for Advanced Proteomics Analyses at the *Institut de la Recherche en Immunologie et Cancérologie* (Montreal, QC), a Node of the Canadian Genomic Innovation Network that is supported by the Canadian Government through Genome Canada. The pCMV-TOM70-2xFLAG-4xUb plasmid was a kind gift from Dr. Xinde Zheng (Salk Institute). The antibodies against MCU, MICU1 and MICU2, as well as the Mfn2 and Mfn1-HA plasmids, were kind gifts from Dr. Heidi McBride. All cell lines and reagents generated by the Montreal Neurological Institute (MNI) IPSC/CRISPR Platform are available without restriction upon request through the Platform under the Open Science Policy of the MNI.

## Additional information

### Funding

| Funder | Grant reference number | Author |
| --- | --- | --- |
| Canadian Institutes of Health Research | Canada Graduate Scholarship | Gian-Luca McLelland |
| Canadian Institutes of Health Research | Foundation Grant | Edward A Fon |

The funders had no role in study design, data collection and interpretation, or the decision to submit the work for publication.

### Author contributions

Gian-Luca McLelland, Conceptualization, Data curation, Investigation, Methodology, Writing—original draft, Project administration, Writing—review and editing; Thomas Goiran, Andrea I Krahn, Investigation; Wei Yi, Investigation, Methodology; Geneviève Dorval, Carol X Chen, Nadine D Lauinger, Sepideh Valimehr, Aleksandar Rakovic, Resources; Isabelle Rouiller, Resources, Supervision; Thomas M Durcan, Resources, Supervision, Writing—review and editing; Jean-François Trempe, Conceptualization, Supervision, Investigation, Methodology, Writing—review and editing; Edward A Fon, Conceptualization, Supervision, Investigation, Methodology, Project administration, Writing—review and editing

### Author ORCIDs

Gian-Luca McLelland http://orcid.org/0000-0002-6122-6252
Edward A Fon http://orcid.org/0000-0002-5520-6239

### Decision letter and Author response
Decision letter https://doi.org/10.7554/eLife.32866.031

Author response https://doi.org/10.7554/eLife.32866.032

# Additional files

## Supplementary files

• Supplementary file 1. MS identification of selective Ub and pUb interactors. Table depicting GST-4xUb interactors that are selective for S65-phosphorylated (top) or unphosphorylated (bottom) Ub. p97-related data (shaded in yellow) are also depicted in *Figure 6—figure supplement 1C*.
DOI: https://doi.org/10.7554/eLife.32866.028

• Transparent reporting form
DOI: https://doi.org/10.7554/eLife.32866.029

## Data availability

All data generated or analysed during this study are included in the manuscript and supporting files. Source data files have been provided for all figures.

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
