## [Decision Letter]

Thank you for submitting your article "Mfn2 ubiquitination by PINK1/parkin gates the p97-dependent release of ER from mitochondria to drive mitophagy" for consideration by *eLife*. Your article has been reviewed by two peer reviewers, and the evaluation has been overseen by Ivan Dikic as the Senior/Reviewing Editor. The reviewers have opted to remain anonymous.

The reviewers have discussed the reviews with one another and the Reviewing Editor has drafted this decision to help you prepare a revised submission.

The author examined ER-mitochondrial contacts (EMCs) during mitophagy. Using quantitative ultrastructural analysis, the authors found that EMCs are disrupted upon mitophagy induction. Focusing on the ER-mitochondrial tether and parkin substrate Mfn2, the authors show that parkin ubiquitinates the HR1 domain of Mfn2 in a PINK1-dependent manner to inhibit the anti-mitophagic property of Mfn2. Furthermore, the authors demonstrated that p97-mediated mitochondrial extraction of ubiquitinated Mfn2 precedes its proteasomal degradation and is required for mitophagy execution. While the authors found that Mfn2 is dispensable for mitochondrial p97 recruitment, McLelland and colleagues revealed that the role of p97 during mitophagy partially depends on Mfn2, possibly through sterically restricting access to other parkin substrates.

Take home message is of broad interest and can reconcile a large amount of contrasting literature into a unifying, testable model. While this work represents an impressive amount of work with experiments performed in wide range of model systems ranging from iPSC-derived neurons to cell-free reconstitution, there a still a few critical issues which needs to be addressed.

Main points:

1) To claim that the CCCP-induced additional Mfn2-positive band on the Mfn2 WB in Figures 1M, 2B, 2C, 6C, 6E and 6F represents an ubiquitinated form of Mfn2, the authors need to show that ubiquitin is covalently attached to Mfn2 by performing IP under denaturing conditions followed by immunoblot for ubiquitin. The same is true for Figure 4E and 4F as well as Figure 8. For assessing which lysine in Mfn2 is required to be ubiquitylated in order to trigger its proteasomal degradation the authors need to monitor Mfn2's ubiquitylation status. Likewise, for the reconstitution of Mfn2 ubiquitination, the authors need to immunoprecipitate Mfn2 from these reactions under denaturing conditions and blot for ubiquitin. Along these lines, the author's lysis buffer used in Figure 2D and 2E does not represent denaturing conditions. Hence, the authors cannot claim that Mfn2 is ubiquitinated/phospho-ubiquitinated. In the current setting, the IP-WB and IP-MS data only shows that ubiquitin and phospho-ubiquitin, are co-purifying with Mfn2. The authors should repeat these experiments in the presence of at least 1% SDS and 10 mM N-ethylmaleimide to support the claim that Mfn2 is covalently modified with ubiquitin or phospho-ubiquitin. The same is true for Figure 6I. Also, it is not clear why the authors do not probe their IP in Figure 2D with the phospho-ubiquitin specific antibody used in Figure 6I. Lastly, the increase of co-purifying phospho-ubiquitin in Figure 6I is not convincing. The author should quantify this phenotype.

2) Figure 2C: The size of the immunoprecipitated Mfn2-positive band in the GFP-parkin IP does not correspond to the CCCP-induced modified Mfn2 band in Figure 2B. The authors should combine their IP conditions with Usp2cc treatment. This would provide actual support of the idea that parkin binds ubiquitinated Mfn2. Moreover, is the parkin-Mfn2 binding stabilized upon proteasome inhibition?

3) The ER-mitochondria EM tethering distance measured here is arbitrarily set at 100 nm; however, the physical tethers measured by Csordas et al., 2006 span 16-25 nm. Authors should reanalyze the EM using two different distances (e.g., 25 and 50 or 100 nm). In addition, because of the morphological alteration of ER and mitochondria, the use of a compound proximity index like the one in Naon et al., 2016 is encouraged.

4) Measurements of mitochondrial membrane potential in cells devoid of Mfns and of ER-mitochondria contacts in the generated Mfn2-/- cell are lacking. They are required to exclude that mitophagy rates are influenced by changes in mitochondrial function (rather than ER-mitochondria proximity) and to confirm that Mfn2 manipulation in their system affects organelle proximity.

---

## [Author Response]

Main points:1) To claim that the CCCP-induced additional Mfn2-positive band on the Mfn2 WB in Figures 1M, 2B, 2C, 6C, 6E and 6F represents an ubiquitinated form of Mfn2, the authors need to show that ubiquitin is covalently attached to Mfn2 by performing IP under denaturing conditions followed by immunoblot for ubiquitin. The same is true for Figure 4E and 4F as well as Figure 8. For assessing which lysine in Mfn2 is required to be ubiquitylated in order to trigger its proteasomal degradation the authors need to monitor Mfn2's ubiquitylation status. Likewise, for the reconstitution of Mfn2 ubiquitination, the authors need to immunoprecipitate Mfn2 from these reactions under denaturing conditions and blot for ubiquitin. Along these lines, the author's lysis buffer used in Figure 2D and 2E does not represent denaturing conditions. Hence, the authors cannot claim that Mfn2 is ubiquitinated/phospho-ubiquitinated. In the current setting, the IP-WB and IP-MS data only shows that ubiquitin and phospho-ubiquitin, are co-purifying with Mfn2. The authors should repeat these experiments in the presence of at least 1% SDS and 10 mM N-ethylmaleimide to support the claim that Mfn2 is covalently modified with ubiquitin or phospho-ubiquitin. The same is true for Figure 6I. Also, it is not clear why the authors do not probe their IP in Figure 2D with the phospho-ubiquitin specific antibody used in Figure 6I. Lastly, the increase of co-purifying phospho-ubiquitin in Figure 6I is not convincing. The author should quantify this phenotype.

We appreciate the reviewer’s concerns regarding the ubiquitination status of Mfn2. We would first like to simply mention that ubiquitination of the mitofusins by parkin during mitophagy was reported some time ago (Tanaka et al., 2010 and other reports) and Mfn2 di-Gly linkages were detected by the Harper group in depolarized cells in a parkin-dependent manner (Sarraf et al., 2013). Indeed, these di-Glys form the basis of the mutagenic analysis performed in Figure 4 and extended in Figure 6. At this point, the field generally accepts that Mfn2 is ubiquitinated by parkin when cells are treated with CCCP (Yamano et al., eLife, 2018 is a very recent example).

That being said, we did seek to solidify our findings according to the reviewer’s suggestions, especially concerning the lysis buffer used in old Figure 2D and E, as well as 6I. We have included the following experiments in the revised manuscript further demonstrating that Mfn2 is both phosphoubiquitinated and covalently linked to Ub:

1) New Figure 2D and E: we have taken GFP-parkin immunoprecipitates from cells treated with CCCP, treated them with the catalytic domain of Usp2, and fractionated the deubiquitination reaction (see new Figure 2D for workflow). We find that Usp2 treatment shifts Mfn2 from the beads (bound to GFP-parkin) to the supernatant. This is consistent with ubiquitinated Mfn2 binding parkin in the IP.

2) New Figure 2H: we have IPed Mfn2 under denaturing conditions, as the reviewer suggested, and blotted for pS65 Ub using the anti-pUb antibody. We detect pUb (and Ub) only in the Mfn2 IP from cells treated with CCCP, consistent with the ubiquitination of this protein.

3) New Figure 2I and J: we have performed a pulldown with the R0RBR module of parkin, which binds pUb with a K_D_ of ~ 20 nM (Sauve et al., 2015). R0RBR WT, but not the A320R mutant (Wauer et al., 2015), robustly pulls down pUb from cell lysates, and we detect both Ub and Mfn2 in the pulldown.

4) New Figure 6G: we have IPed Mfn2 WT and HR1 under denaturing conditions from reconstituted Mfn2 KO cells treated with both CCCP and NMS-873, and show more Ub in the WT IP vs. that of HR1. This result is consistent with Mfn2 HR1 being deficient in its ubiquitination and complimentary to the immunoblot shown in new Figure 6F.

5) We have removed the IP in old Figure 6I. This was complementary to new Figure 6J, which shows that the Mfn2-parkin complex is stabilized by NMS-873. Since this is our focus, we have performed the IP in new Figure 6J three times and now show quantification in new Figure 6K.

We believe that, taken together, these new experiments, in addition to those included in the initial submission of our manuscript, demonstrated that Mfn2 is ubiquitinated by parkin and are in line with what leading groups in the field have observed.

2) Figure 2C: The size of the immunoprecipitated Mfn2-positive band in the GFP-parkin IP does not correspond to the CCCP-induced modified Mfn2 band in Figure 2B. The authors should combine their IP conditions with Usp2cc treatment. This would provide actual support of the idea that parkin binds ubiquitinated Mfn2. Moreover, is the parkin-Mfn2 binding stabilized upon proteasome inhibition?

We thank the reviewer for bringing this point to our attention. Although the migration of Mfn2-Ubn species varies slightly from gel to gel, we have indeed treated GFP-parkin immunoprecipitates with the USP2 catalytic domain (new Figure 2E) as the reviewer suggested. We accordingly observe a decrease in Mfn2 ubiquitination on the beads. Additionally, we have used the parkin R0RBR module, which has nanomolar affinity for pUb (Sauve et al., 2015), to purify pUb conjugates from CCCP-treated cells, and detect ubiquitinated Mfn2 among these species (new Figure 2I and J). These new data demonstrate that parkin binds ubiquitinated Mfn2.

As for the second point: unfortunately, treatment of cells with proteasome inhibitors stabilizes unmodified Mfn2 (see new Figure 1G). In fact, it has been shown that this is due to a maintenance of proteasomal DUB activity while its proteolytic subunits are inhibited (Rakovic et al., 2011). We do, however, achieve the intended result (parkin stabilization) with p97 inhibition (new Figure 6J and K), which acts on Mfn2 upstream or at least simultaneously with the proteasome.

3) The ER-mitochondria EM tethering distance measured here is arbitrarily set at 100 nm; however, the physical tethers measured by Csordas et al., 2006 span 16-25 nm. Authors should reanalyze the EM using two different distances (e.g., 25 and 50 or 100 nm). In addition, because of the morphological alteration of ER and mitochondria, the use of a compound proximity index like the one in Naon et al., 2016 is encouraged.

We have repeated the quantification in Figure 1 using a variety of intermembrane (ER-OMM) distances (<25 nm, <50 nm and <100 nm), as suggested by the reviewer, and this data is shown in new Figure 1—figure supplement 1. We find that, irrespective of the chosen distance, the effect of parkin on ER-OMM connections during mitophagy is the same.

4) Measurements of mitochondrial membrane potential in cells devoid of Mfns and of ER-mitochondria contacts in the generated Mfn2-/- cell are lacking. They are required to exclude that mitophagy rates are influenced by changes in mitochondrial function (rather than ER-mitochondria proximity) and to confirm that Mfn2 manipulation in their system affects organelle proximity.

We have blotted our Mfn2 KO cells for PINK1 – the activator of parkin and the mitophagic cascade – at both the steady-state and during depolarization and show that PINK1 are generally similar under both these conditions in WT and Mfn2 KO cells (new Figure 3—figure supplement 5A). Additionally, we have now provided images showing similar staining in these cell lines with the potentiometric dye TMRM (new Figure 3—figure supplement 4D).